# Detailed analysis of chick optic fissure closure reveals Netrin-1 as an essential mediator of epithelial fusion

Holly Hardy[1], James GD Prendergast[1], Aara Patel[2], Sunit Dutta[3], Violeta Trejo-Reveles[1], Hannah Kroeger[1], Andrea R Yung[4], Lisa V Goodrich[4], Brian Brooks[3], Jane C Sowden[2], Joe Rainger[1]*

[1]The Roslin Institute and Royal (Dick) School of Veterinary Studies, University of Edinburgh, Midlothian, United Kingdom; [2]Birth Defects Research Centre, UCL Great Ormond Street Institute of Child Health, London, United Kingdom; [3]Ophthalmic Genetics and Visual Function Branch, National Eye Institute, National Institutes of Health, Bethesda, United States; [4]Department of Neurobiology, Harvard Medical School, Boston, United States

**Abstract** Epithelial fusion underlies many vital organogenic processes during embryogenesis. Disruptions to these cause a significant number of human birth defects, including ocular coloboma. We provide robust spatial-temporal staging and unique anatomical detail of optic fissure closure (OFC) in the embryonic chick, including evidence for roles of apoptosis and epithelial remodelling. We performed complementary transcriptomic profiling and show that *Netrin*-1 (*NTN1*) is precisely expressed in the chick fissure margin during fusion but is immediately downregulated after fusion. We further provide a combination of protein localisation and phenotypic evidence in chick, humans, mice and zebrafish that Netrin-1 has an evolutionarily conserved and essential requirement for OFC, and is likely to have an important role in palate fusion. Our data suggest that *NTN1* is a strong candidate locus for human coloboma and other multi-system developmental fusion defects, and show that chick OFC is a powerful model for epithelial fusion research.
DOI: https://doi.org/10.7554/eLife.43877.001

*For correspondence:
joe.rainger@roslin.ed.ac.uk

**Competing interests:** The authors declare that no competing interests exist.

## Introduction

Fusion of epithelia is an essential process during normal human development and its dysregulation can result in birth defects affecting the eye, heart, palate, neural tube, and multiple other tissues (*Ray and Niswander, 2012*). These can be highly disabling and are among the most common human birth defects, with prevalence as high as 1 in 500 (*Ray and Niswander, 2012*; *Morrison et al., 2002*; *Nikolopoulou et al., 2017*). Fusion in multiple embryonic contexts displays both confounding differences and significant common mechanistic overlaps (*Ray and Niswander, 2012*). Most causative mutations have been identified in genes encoding transcription factors or signalling molecules that regulate the early events that guide initial patterning and outgrowth of epithelial tissues (*Ray and Niswander, 2012*; *Nikolopoulou et al., 2017*; *Patel and Sowden, 2019*; *Kohli and Kohli, 2012*). However, the true developmental basis of these disorders is more complex and a major challenge remains to fully understand the behaviours of epithelial cells directly involved in the fusion process.

Ocular coloboma (OC) is a structural eye defect that presents as missing tissue or a gap in the iris, ciliary body, choroid, retina and/or optic nerve. It arises from a failure of fusion at the optic fissure (OF; also referred to as the *choroid fissure*) in the ventral region of the embryonic eye cup early in development (*Patel and Sowden, 2019*; *Onwochei et al., 2000*; *Gregory-Evans et al., 2004*).

**eLife digest** Our bodies are made of many different groups of cells, which are arranged into tissues that perform specific roles. As tissues form in the embryo they must adopt precise three-dimensional structures, depending on their position in the body. In many cases this involves two edges of tissue fusing together to prevent gaps being present in the final structure.

In individuals with a condition called ocular coloboma some of the tissues in the eyes fail to merge together correctly, leading to wide gaps that can severely affect vision. There are currently no treatments available for ocular coloboma and in over 70% of patients the cause of the defect is not known. Identifying new genes that control how tissues fuse may help researchers to find what causes this condition and multiple other tissue fusion defects, and establish whether these may be preventable in the future.

Much of what is currently known about how tissues fuse has come from studying mice and zebrafish embryos. Although the extensive genetic tools available in these 'models' have proved very useful, both offer only a limited time window for observing tissues as they fuse, and the regions involved are very small. Chick embryos, on the other hand, are much larger than mouse or zebrafish embryos and are easier to access from within their eggs. This led Hardy et al. to investigate whether the developing chick eye could be a more useful model for studying the precise details of how tissues merge.

Examining chick embryos revealed that tissues in the base of their eyes fuse between five and eight days after the egg had been fertilised, a comparatively long time compared to existing models. Also, many of the genes that Hardy et al. found switched on in chick eyes as the tissues merged had previously been identified as being essential for tissue fusion in humans. However, several new genes were also shown to be involved in the fusing process. For example, Netrin-1 was important for tissues to fuse in the eyes as well as in other regions of the developing embryo.

These findings demonstrate that the chick eye is an excellent new model system to study how tissues fuse in animals. Furthermore, the genes identified by Hardy et al. may help researchers to identify the genetic causes of ocular coloboma and other tissue fusion defects in humans.
DOI: https://doi.org/10.7554/eLife.43877.002

OC is the most common human congenital eye malformation and is a leading cause of childhood blindness that persists throughout life (*Morrison et al., 2002*; *Williamson and FitzPatrick, 2014*). No treatments or preventative measures for coloboma are currently available.

The process of optic fissure closure (OFC) requires the coordinated contributions of various cell types in the fusion environment along the proximal-distal (PD) axis of the ventral eye cup (reviewed in *Patel and Sowden, 2019*; *Onwochei et al., 2000*). In all vertebrates studied so far, these include epithelial cells of both the neural retina (NR) and retinal pigmented epithelium (RPE), and periocular mesenchymal (POM) cells of neural crest origin (*Patel and Sowden, 2019*; *O'Rahilly, 1966*; *Hero, 1990*; *Hero, 1989*; *Gestri et al., 2018*). As the eye cup grows, the fissure margins come into apposition along the PD axis and POM cells are gradually excluded. Through unknown mechanisms, the basal lamina that surround each opposing margin are either breached or dissolved and epithelial cells from each side intercalate and then subsequently reorganise to form a continuum of NR and RPE, complete with a continuous basal lamina. The function, requirement and behaviour of these epithelial cells in the fusing tissue, and their fates after fusion, are not well understood.

Some limited epidemiological evidence suggests environmental factors may contribute to coloboma incidence (*Gregory-Evans et al., 2004*; *Hornby et al., 2003*). However, the disease is largely of genetic origin, with as many 39 monogenic OC-linked loci so far identified in humans and the existence of further candidates is strongly supported by evidence in gene-specific animal models (*Patel and Sowden, 2019*). Most known mutations cause syndromal coloboma, where the eye defect is associated with multiple systemic defects. A common form of syndromal coloboma is CHARGE syndrome (MIM 214800) for which coloboma, choanal atresia, vestibular (inner-ear) and heart fusion defects are defining phenotypic criteria (*Verloes, 2005*). Palate fusion defects and oro-facial-clefting are common additional features of CHARGE (~20% of cases) and in other monogenic syndromal colobomas (e.g. from deleterious mutations in *YAP1*, *MAB21L1*, and *TFAP2A*

[*Rainger et al., 2014*; *Williamson et al., 2014*; *Lin et al., 1992*]), suggestive of common genetic mechanisms and aetiologies, and pleiotropic gene function.

Isolated (i.e. non-syndromal) OC may be associated with microphthalmia (small eye), and the majority of these cases are caused by mutations in a limited number of transcription-factor encoding genes that regulate early eye development (e.g. *PAX6*, *VSX2* and *MAF* [*Patel and Sowden, 2019*; *Williamson and FitzPatrick, 2014*]), implying that abnormal growth of the eye prevents correct OF margin apposition and that fusion defects are a secondary or an indirect phenotype. Indeed, none of these genes have yet been implicated with direct functional roles in epithelial fusion. However, many isolated coloboma cases also exist without microphthalmia, suggesting that in these patients, eye growth occurs normally but the fusion process itself is defective. These OCs are highly genetically heterogeneous and known loci are not recurrent among non-related patients (*Rainger et al., 2017*). Furthermore, despite large-scale sequencing projects, over 70% of all cases remain without a genetic cause identified (*Rainger et al., 2017*).

The most effective and informative models for studying OFC so far have been mouse (*Mus musculus*) and zebrafish (*Danio rerio*). Both have significant experimental advantages, including powerful genetics and robust genomic data. In particular, live-cell imaging with fluorescent zebrafish embryos has proven to be useful in revealing some intricate cell behaviours at the fissure margin during fusion (*Gestri et al., 2018*). However, both models are restrictive for in-depth molecular investigations due to their limited temporal windows of fusion progression and the number of cells actively mediating fusion and subsequent epithelial remodelling.

Here, we present accurate staging and anatomical detail of the process of chick OFC. We show the expansive developmental window of fusion, and the sizable fusion seam available for experimentation and analysis. We take advantage of this to perform transcriptional profiling at key discrete stages during fusion and show significant enrichment for known human OFC genes, and reveal multiple genes not previously associated with OFC. Our analyses also identified specific cellular behaviours at the fusion plate and found that apoptosis was a prominent feature during chick OFC. Furthermore, we reveal Netrin-1 as a mediator of OFC that is essential for normal eye development in evolutionarily diverse vertebrates, and that has a specific requirement during fusion in multiple developmental contexts. This study presents the chick as a powerful model system for further OFC research, provides strong evidence for a novel candidate gene for ocular coloboma, and directly links epithelial fusion processes in the eye with those in broader embryonic tissues.

## Results

### OFC in the chick occurred within a wide spatial and temporal window

The eye is the foremost observable feature in the chick embryo and grows exponentially through development (*Figure 1a*, *Figure 1—figure supplement 1*). The optic fissure margin (OFM) was first identifiable as a non-pigmented region at the ventral aspect of the eye that narrowed markedly in a temporal sequence as the eye increased in size (*Figure 1a*). To gain a clearer overview of gross fissure closure dynamics we first analysed a complete series of resected flat-mounted ventral eye tissue from accurately staged embryos at Hamburger Hamilton stages (HH.St) 25 through to HH.St34 (*n* > 10 per stage; *Figure 1—figure supplement 1*). The OFM was positioned along the proximal-distal (P-D) axis of the eye, from the pupillary (or collar) region of the iris to the optic nerve. Progressive narrowing of the OFM was observed between HH.St27 to HH.St31, characterised by the appearance of fused OFM in the midline that separated the non-pigmented iris from the posterior OFM (*Figure 1—figure supplement 1*). Both these latter regions remained unpigmented throughout development and we found they were associated, respectively, with the development of the optic nerve and the pecten oculi - a homeostasis-mediating structure that extends out into the vitreous from the optic nerve head and is embedded in the proximal OFM (*Figure 1—figure supplements 1* and *2*) (*Wisely et al., 2017*). The distal region of the pecten was attached to blood vessels that invade the eye globe through the open region of the iris OFM. This iris region remained open throughout development and well after hatching (*Figure 1—figure supplement 2*). A recent study reported that the proximal chick OFM closes via the intercalation of incoming astrocytes and the outgoing optic nerve (*Bernstein et al., 2018*), in a process that does not reflect the epithelial fusion seen during human OFC (e.g. mediated by epithelial cells of the RPE and neural retina)

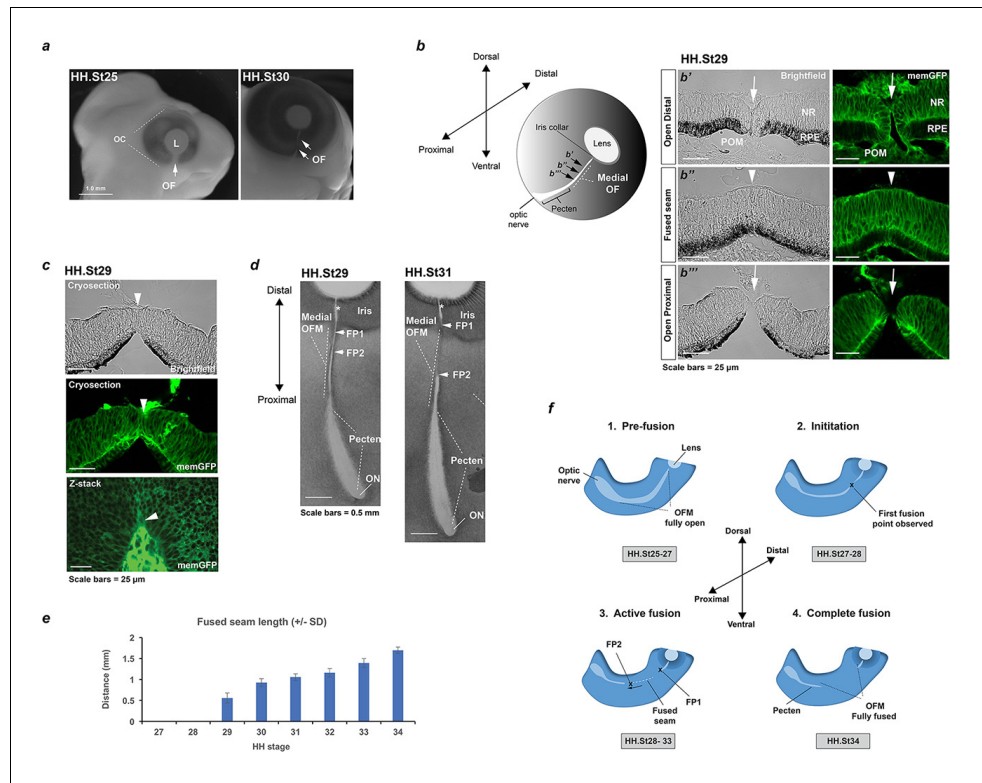

**Figure 1.** Analysis of chick optic fissure closure. (**a**) Chicken embryos at HH.St25 and HH.St30 illustrated the optic fissure (OF; arrows) as a non-pigmented region in the ventral aspect of the developing eye. (**b**) *Left*: Schematic showing orientation of the developing chick optic fissure with respect to the whole embryonic eye. Dorsal-ventral and proximal-distal axes are indicated. This study focused on the medial optic fissure (marked by white hatching) distal to the developing pecten and optic nerve. *Right*: brightfield and fluorescent confocal microscopy using memGFP cryosections illustrated the open (arrow) and fused seam (arrowhead) regions in chick OFM. The location and planes of the cut sections along the D-P axis are indicated in the accompanying schematic. (**c**) Brightfield and fluorescent confocal microscopy of memGFP OFM sections unambiguously defined the location of fusion plates (arrowheads, top and middle panels) at all stages throughout OFC, combined with flat-mounted memGFPs. Bottom panel: representative single plane confocal z-stack projection image clearly indicated FP2. (**d**) Brightfield microscopy of flat-mounted ventral eyes revealed the tissue dynamics during closure and coinciding with location of fusion plates (FPs). At HH.St29 the medial OFM had narrowed markedly along the P-D axis between the iris and the proximal region, with FP1 and FP2 (arrowheads) closely positioned in the distal OF. At HH.St31 the medial OFM had become fully pigmented in the fused seam, and the distance between FP1 and FP2 (arrowheads) had lengthened in the P-D axis. An opening remained in the OFM at the iris region (asterisk). (**e**) Histogram to illustrate fused seam length at each HH stage (error bars = s.d.). Quantitative data of OFM progression obtained from flat mounts and cryosections are provided in *Table 1*. (**f**) Schematic representation of chick OFC progression in the distal and medial retina. *1. Pre-fusion*: A fully open OFM is evident in the ventral retina at stages HH.St25-27; *2. Initiation*: At HH.St27-28 the first fused region is observed in the distal-medial OFM; *3. Active fusion*: fusion extends briefly in the distal direction but then stops in the presumptive iris to leave an open region throughout development. Fusion proceeds markedly proximally with FP2 extending towards the pecten. *4. Complete fusion*: Fusion stops proximally when FP2 meets the fused pecten region. The fusion seam is complete with a complete continuum of both NR and RPE layers in the ventral eye. <u>Abbreviations</u>: L, lens; OC, optic cup, OF, optic fissure; ON, optic nerve; FP, fusion plate; HH, Hamburger Hamilton staging; RPE, retinal pigmented epithelia; NR, neural retina; POM, periocular mesenchyme.

DOI: https://doi.org/10.7554/eLife.43877.003

The following figure supplements are available for figure 1:

**Figure supplement 1.** Anatomical and histological survey of chick OFC.
DOI: https://doi.org/10.7554/eLife.43877.004
**Figure supplement 2.** Anatomical features of iris and pecten in relation to OFC in the chick eye.
DOI: https://doi.org/10.7554/eLife.43877.005

(*O'Rahilly, 1966*; *Bernstein et al., 2018*). To assess the utility of the chick as a model for human OFC and epithelial fusion, we therefore focused our study on OFC progression in the distal and medial eye.

Using serial sections from memGFP (*Rozbicki et al., 2015*) and wild-type embryos, we then unambiguously identified open fissure and fused seam regions of the medial-distal OFM (*Figure 1b*). The fused seams were defined by epithelial continuum in both the developing retinal pigmented epithelia (RPE) and neural retina (NR) layers. We also identified the *fusion plates* undergoing active fusion using sections and z-stack confocal microscopy (*Figure 1c*). Serial sectioning at stages HH.St25-34 provided qualitative data for the identification of fusion plates during the progression of chick OFC (*Table 1*). We then combined these data with fusion seam length measurements taken from flat mounted fissures to provide a robust quantitative framework of fusion progression (*Table 2*). In all analyses, we observed no evidence for fusion in the medial or distal OFM at stages before HH.St27 (*Figure 1*; *Figure 1—figure supplement 1*; *Table 1*). Fusion was first initiated between HH.St27-28 as confirmed by the definitive appearance of joined epithelial margins at a single fusion point (FP). By HH.St29, the fused area had expanded to generate a fused seam of 0.56 mm (SD: ± 0.12 mm; *Figure 1d–e*) with two fusion plates, FP1 and FP2 at the distal and proximal limits, respectively. The position of FP1 became fixed at approximately 0.5 mm (SD: ± 0.04 mm) from the developing pupillary region of the iris in all subsequent developmental stages (*Table 2*, $n = 60$ fissures analysed), and the region between FP1 and the iris remained fully open throughout ocular development (*Figure 1—figure supplement 2* and *Table 1*). In contrast, the location of FP2 became progressively more proximal until HH.St34 (*Table 2*), when FP2 was no longer distinguishable from the pecten (by flat mount or cryosections). This total expansion created a fused epithelial seam of ~1.7 mm at its maximum length (SD: ± 0.07 mm, *Figure 1e*). In summary, we observed four distinct phases of fusion (*Figure 1f*): (1) *pre-fusion* when the entire OFM is open (up to HH.St27); (2) *fusion initiation* at HH.St27-28 in the medial OFM with the appearance of a single medial FP; (3) *active fusion* as two FPs separate with the expansion of a fused seam along the P-D axis (HH.St29-33); and (4) *complete fusion* as the entire OFM is fully fused in the medial OFM (by HH.St34). The process is active between HH.St27-HH.St34 and proceeds over ~66 hr.

## Chick OFC was characterised by the breakdown of basement membranes, loss of epithelial morphology and localised apoptosis

By defining fusion progression and the location of the fusion plates during chick OFC, we could then accurately assess the cellular environment within these regions. Immunostaining for the basement membrane (BM) (or basal lamina) marker Laminin-B1 on cryo-sectioned fissure margins (*Figure 2a*) indicated that fusion occured between cells of the RPE and neural retinal, as observed in human OFC (*O'Rahilly, 1966*). Fusion between opposing margins was defined by a reduction of Laminin-B1

**Table 1.** Qualitative analysis of fusion plates observed per developmental stage by cryosections and H and E.

| HH stage | Fusion plates identified | | N per stage |
|---|---|---|---|
| | **1x FP only** | **Both FP1 and FP2** | |
| 25 | 0 | 0 | 4 |
| 26 | 0 | 0 | 4 |
| 27 | 1 | 0 | 3 |
| 28 | 3 | 1 | 4 |
| 29 | 1 | 4 | 5 |
| 30 | 0 | 4 | 4 |
| 31 | 0 | 3 | 3 |
| 32 | 0 | 5 | 5 |
| 33 | 1 | 2 | 3 |
| 34 | 3 | 0 | 3 |

DOI: https://doi.org/10.7554/eLife.43877.006

**Table 2.** Quantitative measurements of key features during OFC progression using flat mounted WT and mem-GFP fissures.
Total OFM length includes optic nerve and pecten. * Fused fissures observed were too small to measure accurately (<0.1 mm).

| HH stage | Mean total OFM length (mm) | ± SD | Mean length of fused seam (mm) | ± SD |
|---|---|---|---|---|
| 27 | 2.20 | 0.15 | - | - |
| 28 | 2.92 | 0.33 | * | * |
| 29 | 3.58 | 0.28 | 0.56 | 0.12 |
| 30 | 4.38 | 0.17 | 0.93 | 0.09 |
| 31 | 4.50 | 0.25 | 1.09 | 0.13 |
| 32 | 4.77 | 0.16 | 1.15 | 0.10 |
| 33 | 5.31 | 0.23 | 1.39 | 0.10 |
| 34 | 5.67 | 0.16 | 1.70 | 0.07 |

DOI: https://doi.org/10.7554/eLife.43877.007

at the edges of the directly apposed fissures, followed the appearance of a continuum of BM overlying the basal aspect of the neural retina. Periocular mesenchymal cells were removed from between the fissure margins as fusion progressed. Using a histological approach, we then provided evidence that both the RPE and NR directly contribute cells to the fusion plate (*Figure 2b*). We also observed that within the fusion plates there was marked epithelial remodelling of both cell types, beginning after apposition of the OFM edges. In contrast, at the fused seam we observed NR and RPE cells were realigned into apical-basal orientation and were indistinguishable from regions outside of the OFM, indicating that the fusion process was complete.

To determine whether the expanding seam between FP1 and FP2 was a result of active directional fusion (e.g. 'zippering'), or was driven by localised cell-proliferation within the OFM seam (e.g. pushing forward static fusion plates), we used phospho-Histone-H3A (PH3A) as a marker for S-phase nuclei in mitotic cells and revealed there was no significant enrichment within the fusion seam (*Figure 2—figure supplement 1*). These results suggested that localised cell-proliferation within the seam was not a major mechanism for seam expansion during chick OFC, and further work is required to elucidate the precise mechanisms that drive seam expansion. We then sought to establish whether axonal ingression was a feature of chick OFC in the distal-medial OFM. Using Neurofilament-145 immunofluorescence, we found a complete absence of axonal processes in open, fusing, and fused regions of the distal-medial chick OFM (*Figure 2—figure supplement 2*). In contrast, at the same stages we found marked enrichment for axons within the proximal OFM and pecten region, providing further evidence that these regions of the chick optic fissure are distinct (*Bernstein et al., 2018*).

Programmed-cell death has been previously associated with epithelial fusion in multiple developmental contexts but the exact requirements for this process remain controversial (*Ray and Niswander, 2012*). Even within the same tissues differences arise between species - for example, apoptotic cells are clearly observed at the mouse fusion plate during OFC (*Hero, 1990*) but are not routinely found in zebrafish (*Gestri et al., 2018*). We therefore asked whether apoptosis was a major feature of chick OFC. Using HH.St30 eyes undergoing active fusion, we performed immunofluorescence staining for the pro-apoptotic marker activated Caspase-3. We consistently identified apoptotic foci within RPE and NR at both fusion plates, in the adjacent open fissure margin, and at the nascently fused seam with both cryo-section and whole-mount samples (*Figure 2c*; *Figure 2—figure supplement 2*). Foci were not found consistently in other regions of the eye or ventral retina (not shown). By quantifying the number of positive A-Casp-3 foci at FP2, we found that apoptosis was specifically enriched in the active fusion environment but was absent from fused seam >120 μm and from open regions > 250 μm beyond FP2 (*Figure 2d*), indicating that apoptosis is a specific feature of OFC in the chick eye.

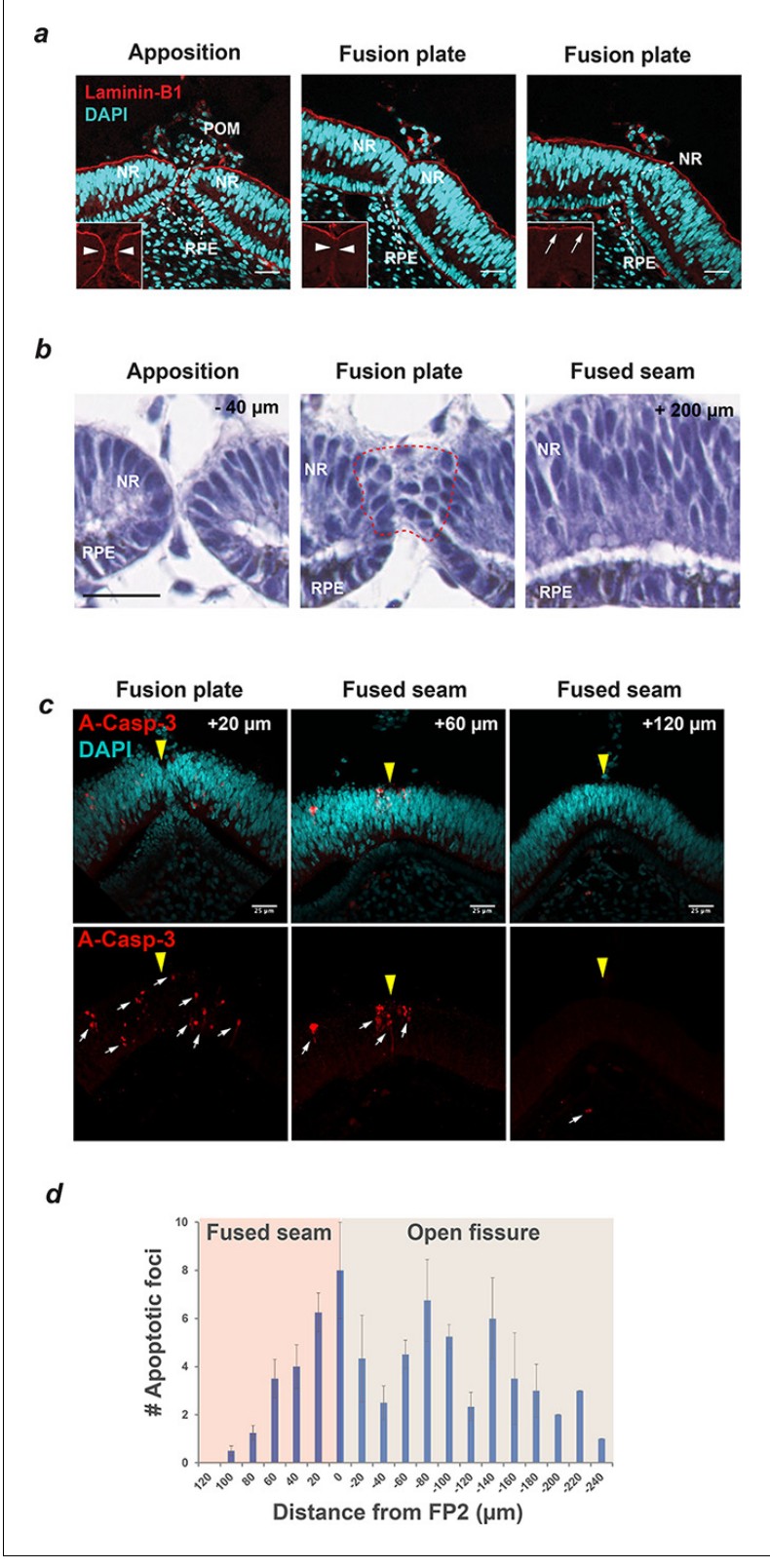

**Figure 2.** Basement membrane remodelling, loss of epithelial characteristics and apoptosis are defining features of Chick OFC. (a) Immunostaining for the basement membrane (BM) component Laminin-B1 and nuclear staining (DAPI) using confocal microscopy illustrated that fusion was preceded by the dissolution of BM (compare arrowheads in boxes) as the fissure margins came into contact at the fusion plate, and that fusion was

*Figure 2 continued on next page*

*Figure 2 continued*

characterised by the generation of a BM continuum at the basal aspect of the neural retina (arrows). Nuclear staining indicated that cells of the retinal pigmented epithelium (RPE) and neural retina (NR) contributed to the fusion plate and that periocular mesenchymal cells were removed from the region between the apposed margins. Images are from a single OFM and are representative of n ≥ 3 samples. (**b**) H and E staining on paraffin sections at FP2 showed apposed fissure margins with well organised epithelia in NR and RPE (−40 μm from FP2); subsequent sections at the fusion plate showed loss of epithelial organisation in both cell types (within hatching); at the fused seam (+200 μm from FP2) continuous organised layers were observed in both NR and RPE epithelia. Note that fusion occurred from contributions of both NR and RPE. (**c**) Immunostaining for the apoptosis marker activated Caspase-3 (A-Casp3) on serial cryo-sectioned OFMs (HH.St30) using confocal microscopy (z-stack projections) indicated that A-Casp3 positive foci (arrows) were enriched in epithelia at the OFM and in the nascently fused seam. The midline OFM, including the fusion points, is indicated by yellow arrowheads in all panels. OFMs were counterstained with DAPI. (**d**) Quantitation of A-Casp3 foci from serially-sectioned OFMs confirmed significant enrichment at FP2, with a graded reduction in apoptotic cells in both directions away from the fusion plate. Data shown are the mean of all measurements (*n* = 4); error bars = 95% Confidence intervals. Scale bars = 25 μm in **a** and **c**, =20 μm in **b**.

DOI: https://doi.org/10.7554/eLife.43877.008

The following figure supplements are available for figure 2:

**Figure supplement 1.** Analysis of proliferation in the OF margin.

DOI: https://doi.org/10.7554/eLife.43877.009

**Figure supplement 2.** Analysis of axonal processes and apoptosis during chick OFC.

DOI: https://doi.org/10.7554/eLife.43877.010

## Transcriptional profiling revealed genetic conservation between chick and human OFC

We took advantage of the size and accessibility of the embryonic chick eye to perform transcriptomic profiling with the objectives of: (i) assessing the utility of the chick as a genetic model for human OFC by expression of chick orthologues for known disease genes; and (ii) to identify novel genes that are required for OFC. Using HH.st25-26 eyes (pre-fusion; approx. embryonic day E5), segmental micro-dissection of the embryonic chick eye was first performed to obtain separate OFM, ventral eye, dorsal eye and whole eye samples (*Figure 3—figure supplement 1*). We took care to not extract tissue from the pecten or optic nerve region of the developing OFM to ensure we obtained transcriptional data for the distal and medial OFM only. Cognate tissues were pooled, RNA was extracted, and region-specific transcriptomes were determined using total RNAseq and analysed to compare mean transcripts per million (TPM) values (*Figure 3—source data 1*). Pseudoalignment to the Ensembl chicken transcriptome identified 30,265 expressed transcripts across all tissue types. To test whether this approach was sensitive enough to reveal domain-specific expression in the developing chick eye, we compared our RNAseq expression data for a panel of genes with clear regional specific expression from a previous study of mRNA in situ analyses in the early developing chick eye cup (*Peters and Cepko, 2002*). Markers of the early dorsal retina (*Efnb1, Efnb2, Vsx2, Tbx5, Aldh1A1*) clustered as dorsal-specific in our RNAseq data, whereas known ventral markers (*Crx, Maf1, Pax2, Aldh6 [Ald1a3], Vax1, and Rax1*) were strongly expressed in our fissure and ventral transcriptomes (*Figure 3—figure supplement 1*), which validated this approach to reveal OFC candidate genes.

We then repeated the analysis, collecting OFM, ventral tissue and whole eye and included stages HH.st27-28 (~E6; during initiation) and HH.st28-30 (~E7; during active fusion) as discrete time-points (*Figure 3—figure supplement 1*). Dorsal tissue was not extracted for these stages. Correlation matrices for total transcriptomes of each sample indicated one of the HH.st25-26 fissure samples as an outlier, but otherwise that there was close correlation between all the other samples (Pearson's correlation coefficient >0.9; *Figure 3—figure supplement 1*). Quantitative analyses identified 14,262 upregulated genes and 14,125 downregulated genes in the fissure margin at the three time points (*Figure 3a*; fissure versus whole eye. False discovery rate (FDR) adjusted p-value<0.05. The largest proportion of these differential expressed genes (DEGs) were observed at HH.st25-27, most likely reflecting the periocular tissue between the fissure margins. Remarkably few DEGs were shared between stages. We used fold change (FC) analysis to identify biologically-relevant

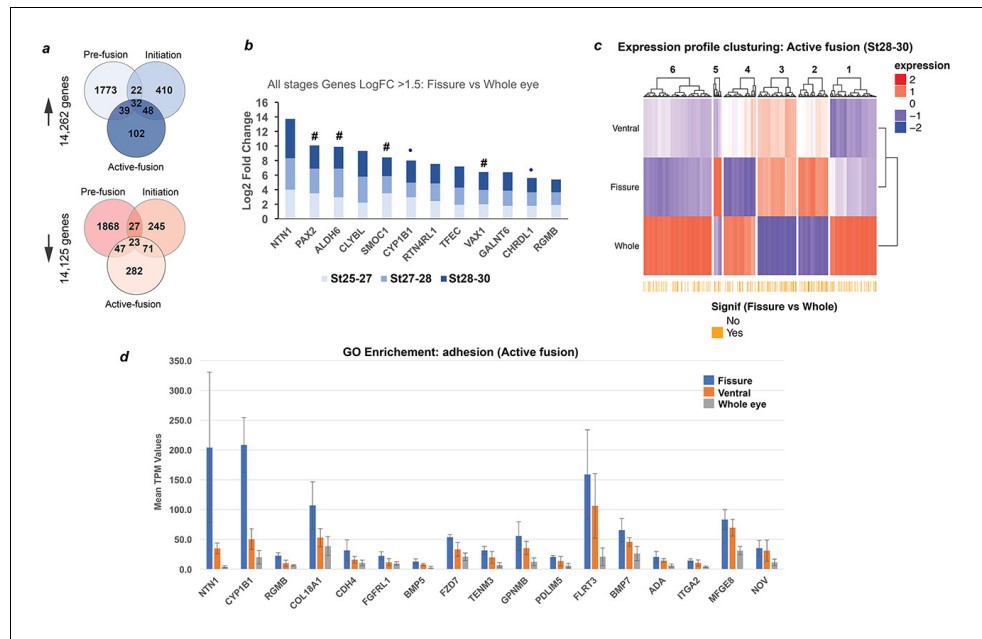

**Figure 3.** Transcriptional profiling in chick optic fissure closure. (**a**) Transcriptional profiling using microdissected regions of the developing chick eye at E5 (HH.St25-27; pre-fusion), E6 (HH.St27-28; initiation), and E7 (HH.St28-30; during active fusion) revealed multiple DEGs at each stage. (**b**) *NTN1* was the highest expressing gene of 12 fissure-specific DEGs (fissure vs whole eye) throughout all stages of chick OFC (Log2 FC >1.5; FDR < 0.05). These included the known human coloboma associated genes (indicated by #): *SMOC1, PAX2, VAX1* and *ALDH6,* in addition to the coloboma candidates from other animal studies *CHRDL1* and *CYP1B1* (indicated by •). (**c**) Clustering for relative expression levels at active fusion stages (HH.St28-30) revealed three independent clusters (2, 3, and 5) where expression levels trended with Fissure >ventral > whole eye. (**d**) Analysis of normalised mean expression values (TPM, n = 3 technical replicates; error bars = 1 x standard deviation) from clusters 2, 3, five at HH.St28-30 for the Gene Ontology enriched pathways (p<0.0001; Biological fusion [GO:0022610], and Epithelial fusion [GO:0022610]) revealed significant fissure-specific expression for highly expressed (TPM >100) genes - *NTN1, FLRT3, CYP1B1* and *COL18A1* - in addition to other potential candidate genes for roles in OFC. *NTN1* (TPM >200) was the highest expressed fissure-specific DEG identified during active fusion.
DOI: https://doi.org/10.7554/eLife.43877.011

The following source data and figure supplement are available for figure 3:

**Source data 1.** Kallisto analysis of RNAseq data from segmentally dissected HH.St25-26/E5 chick eyes.
DOI: https://doi.org/10.7554/eLife.43877.013

**Source data 2.** Limma analysis of RNAseq data from segmentally dissected chick eyes at all stages.
DOI: https://doi.org/10.7554/eLife.43877.014

**Figure supplement 1.** Schema and validation data for transcriptional profiling during OFC.
DOI: https://doi.org/10.7554/eLife.43877.012

differential gene expression (Log$_2$FC $\geq$1.5 or $\leq$−1) in the fissure compared to whole eye, we found 1613, 2971 and 1491 DEGs at pre-fusion, initiation, and active fusion, respectively (***Figure 3—source data 2***). Refining our analysis to identify only those DEGs common across all stages revealed 12 genes with increased expression in the fissure and 26 with decreased expression (***Figure 3b***; ***Table 3***). Of these upregulated fissure-specific genes, causative mutations have previously been identified in orthologues of *PAX2, SMOC1, ALDH1A3,* and *VAX1* in human patients with coloboma or structural eye malformations (***Patel and Sowden, 2019***; ***Williamson and FitzPatrick, 2014***), and some of these genes, such as PAX2 and inhibitors of BMP expression, induce coloboma phenotypes when overexpressed in the developing ventral chick eye (***Gregory-Evans et al., 2004***; ***Sehgal et al., 2008***). In addition, targeted manipulations of orthologues of both *CHRDL1* and *CYP1B1* have recently been shown to cause coloboma phenotypes in *Xenopus* and *zebrafish*, respectively (***Pfirrmann et al., 2015***; ***Williams et al., 2017***). The remaining fissure-specific genes (*NTN1,*

**Table 3.** Fissure-Specific Differentially expressed genes (q < 0.05; LogFC:≥1.5 and ≤−1) at all stages analysed. Genes with increased expression are depicted in grey.

| ENSEMBL ID | HGNC ID | LogFC: Fissure vs whole (HH.St25-27)~E5 | FDR adjusted P value | LogFC: Fissure vs whole (HH.St27-28)~E6 | FDR adjusted P value | LogFC: Fissure vs whole (HH.St28-30)~E7 | FDR adjusted P value |
|---|---|---|---|---|---|---|---|
| ENSGALG00000023626 | NTN1 | 3.98 | 5.11E-05 | 4.34 | 8.16E-05 | 5.41 | 3.06E-07 |
| ENSGALG00000005689 | PAX2 | 3.48 | 9.36E-06 | 3.41 | 2.11E-05 | 3.18 | 4.14E-06 |
| ENSGALG00000033365 | ALDH6 | 2.97 | 1.00E-05 | 3.94 | 1.75E-04 | 3.00 | 4.91E-05 |
| ENSGALG00000016875 | novel gene | 2.21 | 4.57E-05 | 3.56 | 8.72E-07 | 3.55 | 2.62E-08 |
| ENSGALG00000009415 | SMOC1 | 3.49 | 1.46E-05 | 2.36 | 3.92E-03 | 2.60 | 8.93E-05 |
| ENSGALG00000025822 | CYP1B1 | 2.95 | 1.11E-05 | 2.03 | 1.34E-02 | 3.02 | 1.55E-05 |
| ENSGALG00000021589 | RTN4RL1 | 2.41 | 5.79E-03 | 2.43 | 8.00E-03 | 2.69 | 4.22E-04 |
| ENSGALG00000040557 | TFEC | 1.93 | 8.70E-03 | 2.35 | 5.01E-03 | 2.90 | 9.86E-04 |
| ENSGALG00000009261 | VAX1 | 1.99 | 6.95E-04 | 1.96 | 3.15E-03 | 2.49 | 1.55E-05 |
| ENSGALG00000041101 | GALNT6 | 1.78 | 3.45E-04 | 2.07 | 6.97E-04 | 2.55 | 6.85E-06 |
| ENSGALG00000008072 | CHRDL1 | 1.79 | 3.85E-05 | 1.86 | 1.03E-03 | 1.94 | 2.49E-05 |
| ENSGALG00000015284 | RGMB | 1.89 | 1.37E-02 | 1.73 | 2.43E-02 | 1.76 | 7.63E-03 |
| ENSGALG00000011413 | novel gene | −1.53 | 1.13E-02 | −1.34 | 3.34E-02 | −1.69 | 1.60E-02 |
| ENSGALG00000004270 | ALDH1A2 | −1.20 | 4.06E-02 | −1.80 | 9.46E-03 | −1.76 | 2.74E-03 |
| ENSGALG00000010801 | TMEM61 | −2.07 | 8.63E-03 | −1.49 | 3.77E-02 | −1.93 | 5.90E-03 |
| ENSGALG00000003842 | GHRH | −1.33 | 4.58E-02 | −2.60 | 1.48E-02 | −2.62 | 5.28E-03 |
| ENSGALG00000012712 | RBM24 | −2.57 | 9.39E-04 | −2.00 | 1.69E-02 | −2.35 | 2.80E-03 |
| ENSGALG00000012644 | novel gene | −1.85 | 4.91E-03 | −2.58 | 1.38E-02 | −3.18 | 9.33E-04 |
| ENSGALG00000003324 | PRRX1 | −1.52 | 4.65E-02 | −2.77 | 2.21E-02 | −3.42 | 1.32E-03 |
| ENSGALG00000007706 | FGF8 | −2.20 | 2.94E-03 | −3.10 | 8.72E-04 | −2.64 | 9.86E-04 |
| ENSGALG00000010929 | SPARCL1 | −3.16 | 3.03E-03 | −1.77 | 4.19E-02 | −3.17 | 6.48E-04 |
| ENSGALG00000034585 | CP49 | −3.65 | 6.32E-06 | −1.93 | 1.55E-02 | −2.59 | 3.60E-04 |
| ENSGALG00000038848 | MSX2 | −2.19 | 4.15E-03 | −3.35 | 1.15E-02 | −2.92 | 5.01E-03 |
| ENSGALG00000004279 | GRIFIN | −3.97 | 7.94E-04 | −2.71 | 2.55E-02 | −1.92 | 4.89E-02 |
| ENSGALG00000004569 | UNC5B | −1.41 | 4.81E-03 | −4.14 | 4.56E-08 | −3.21 | 2.62E-08 |
| ENSGALG00000019802 | novel gene | −2.24 | 1.56E-02 | −3.43 | 4.36E-02 | −3.59 | 9.52E-03 |
| ENSGALG00000043175 | novel gene | −3.59 | 7.36E-03 | −2.99 | 3.27E-02 | −2.91 | 2.59E-02 |
| ENSGALG00000005613 | novel gene | −2.96 | 6.50E-04 | −2.21 | 1.99E-02 | −4.40 | 2.06E-04 |
| ENSGALG00000015015 | CYTL1 | −2.43 | 3.39E-02 | −3.13 | 4.74E-02 | −5.14 | 5.01E-03 |
| ENSGALG00000004035 | CRYBA1 | −5.04 | 1.21E-04 | −2.56 | 1.95E-02 | −3.33 | 2.00E-03 |
| ENSGALG00000006189 | CRYGN | −4.66 | 6.22E-04 | −4.25 | 1.82E-02 | −4.97 | 9.33E-04 |
| ENSGALG00000012470 | LYPD6 | −2.49 | 1.20E-02 | −4.64 | 6.97E-04 | −7.13 | 5.09E-06 |
| ENSGALG00000008253 | TBX5 | −3.50 | 3.48E-04 | −6.73 | 5.98E-04 | −4.39 | 6.02E-05 |
| ENSGALG00000015147 | ALDH1A1 | −5.06 | 1.46E-05 | −4.96 | 1.22E-04 | −4.79 | 1.55E-05 |
| ENSGALG00000042119 | MIP | −4.47 | 2.10E-03 | −5.43 | 3.97E-02 | −6.15 | 3.54E-03 |
| ENSGALG00000005634 | CRYBA4 | −5.47 | 2.65E-04 | −4.94 | 1.61E-02 | −7.17 | 6.56E-04 |
| ENSGALG00000005630 | CRYBB1 | −5.36 | 1.72E-04 | −6.97 | 4.56E-03 | −6.24 | 1.37E-04 |
| ENSGALG00000008735 | BFSP1 | −6.48 | 5.53E-04 | −6.23 | 1.76E-02 | −8.63 | 1.97E-03 |

DOI: https://doi.org/10.7554/eLife.43877.015

*RTN4RL1*, *TFEC*, *GALNT6*, CLYBL and *RGMB*) had not been previously associated with OFC defects to the best of our knowledge.

## Clustering analysis revealed *NTN1* as a fusion-specific OFC gene

Clustering for relative expression levels of the RNAseq data at active fusion stages (HH.St28-30) revealed three independent clusters (2, 3, and 5) where expression profiles matched Fissure >ventral > whole eye (*Figure 3c*). We hypothesised that analysis of these clusters would reveal genes with fusion-specific functions during OFC. Of the three clusters with this profile, ontology analyses showed significant enrichment for sensory organ development and eye development processes (*FDR q* < 0.001, 10 genes) and for adhesion processes (*Figure 3—figure supplement 1*; *FDR q* < 0.05, 25 genes; Biological adhesion [GO:0022610] and cell adhesion [GO:0022610]), of which 17 genes had mean TPM values > 10. Within this group, multiple candidates for roles during OFC fusion were revealed, such as several transmembrane proteins, Integrin-A2, Cadherin-4, Collagen 18A1 and FLRT3 (*Figure 3d*). However, of these *NTN1* was the highest expressed and most fissure-specific (mean TPM values: Fissure = 204; ventral = 35; and whole eye = 4).

## *Netrin-1* was specifically and dynamically expressed in the fusing OFM

We used RNAscope, colorimetric in situ hybridisation, and immunostaining with NTN1-specific antibodies to determine the precise location of Netrin-1 in the chick eye (*Figure 4* and *Figure 4—figure supplement 1*). We observed highly specific expression in both neuroepithelial retina and RPE cells at the fissure margins during active fusion at HH.St29-30 (*Figure 4a*). This was consistent at both fusion plates (FP1 and FP2), and in both locations *NTN1* expression was markedly reduced in the fused seam compared to expression in the adjacent open margins. Immunofluorescence revealed that, consistent with *NTN1* mRNA, NTN1 protein was specifically localised to the basal lamina at the opposing edges of the OFM, and to both RPE and neuroepithelial retina cells in this region (*Figure 4b–c*, *Figure 4—figure supplement 1*). To test the significance of our findings to other vertebrates, we first asked whether this localisation was conserved to the human OFM. Immunofluorescence analysis for NTN1 (hNTN1) in human embryonic fissures during fusion stages (Carnegie Stage CS17) displayed remarkable overlap with our observations in chick, with protein signal localised specifically to open and fusion plate regions of OFM at the NR and RPE (*Figure 4d*), and an absence of hNTN1 in fused seam. Consistent with the protein localisation, RNAseq analysis on laser-captured human fissure tissue showed a 32x fold increase in *hNTN1* expression compared to dorsal eye (Patel and Sowden; *manuscript in preparation*). Microarray analyses had previously observed enrichment for *Ntn1* in the mouse fissure during closure stages (*Brown et al., 2009*), so we then analysed Ntn1 protein localisation in equivalent tissues in the mouse optic fissure (fusion occurs around embryonic day E11.5 and is mostly complete by E12.5 (*Hero, 1990*). We observed consistency in both cell-type and positional localisation of Ntn1 protein (*Figure 4e*), and that Ntn1 protein was not detected in the fused seam at E12.5 (immunoreactivity for NTN1 was observed in the proximal optic nerve region at this stage; *Figure 4—figure supplement 2*).

## Complete loss of netrin caused coloboma and multisystem fusion defects in vertebrates

Our results suggested that Netrin-1 has an evolutionarily conserved role in OFC and prompted us to test if NTN1 is essential for this process. We therefore analysed mouse embryos of WT and Netrin-null (*Ntn1^-/-^*; *Yung et al., 2015*) littermates at embryonic stages after OFC completion (E15.5-E16.5) (*Hero, 1990*) and observed highly penetrant ocular coloboma in *Ntn1^-/-^* mutants (>90%; *n* = 10/11; *Figure 4f*). Mutant eyes analysed at earlier stages of eye development (E11.5) when fusion is first initiated (*Hero, 1990*) were normal (*n* = 4 *Ntn1^-/-^* embryos; 8x eyes analysed in total), with fissure margins positioned directly in appositional contact each other (*Figure 4—figure supplement 2*). We also observed variably penetrant orofacial and palate fusion defects in mutant mice (*Figure 4g*;~36%; n = 4/11 *Ntn1^-/-^* embryos), indicating that NTN1 may also have an important role in fusion during palatogenesis and craniofacial development.

Finally, we then tested whether Netrin deficiency would cause similar ocular defects in other vertebrates and generated germline *netrin-1* mutant zebrafish by creating a nonsense mutation in the first exon of *ntn1a* using CRISPR/Cas9 gene editing (*Figure 4—figure supplement 3*). We inter-

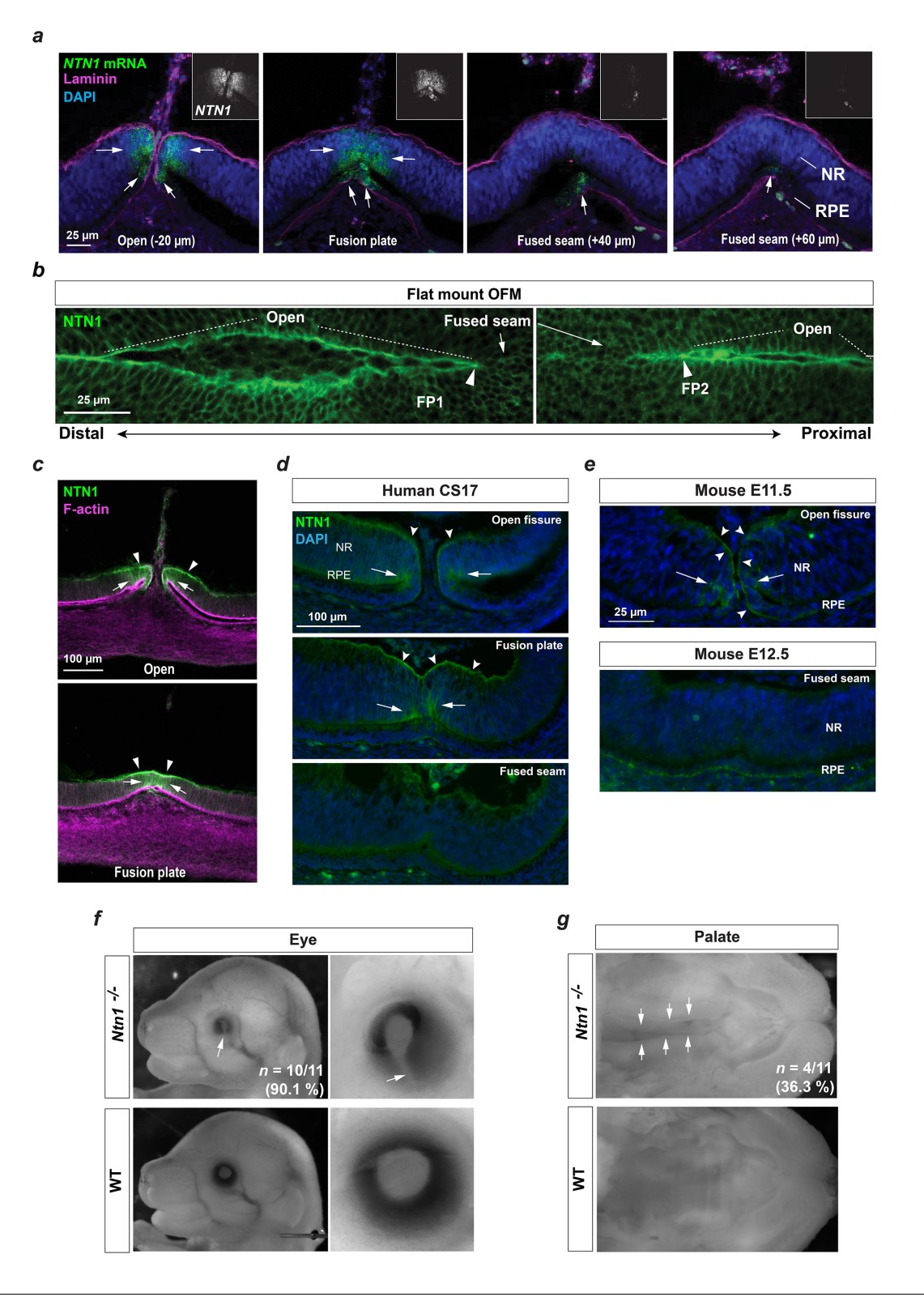

**Figure 4.** A conserved fusion-specific requirement for NTN1 in OFC and palate development. (a) RNAscope analysis of *NTN1* mRNA (green, and grey in insets) in HH.St29 OFMs revealed fissure-specific *NTN1* expression (arrows) with strongest signal observed at open regions and in the fusion plate, and reduced in the adjacently fused seam. *NTN1* expression was localised to cells of both the NR and RPE. Fusion progression was indicated using anti-laminin co-immunofluorescence (magenta). Images shown are maximum intensity projections of confocal Z-stacks. (b) Single-plane

*Figure 4 continued on next page*

*Figure 4 continued*

confocal images of immunofluorescence analysis for NTN1 on flat-mounted distal (FP1) and proximal (FP2) OFM revealed enriched protein localisation at the edges of the open fissure margins and reduced in the fused seam. (**c**) Immunostaining on cryosectioned OFM at the open and fusion plate at HH.St29 revealed NTN1 was specifically localised to the basal lamina (arrowheads) and to the epithelia of the neural retina and RPE (arrows) at the OFM. (**d**) Immunostaining on CS17 human foetal eye sections revealed human Netrin-1 (hNTN1) was localised to NR epithelia (arrows) and at the overlying basal lamina (dented arrowheads) at the fissure margins. hNTN1 was absent from the fused seam epithelia. (**e**) Immunostaining for mouse Netrin-1 (mNtn1) in during active fusion stages (E11.5) showed mNtn1 was localised at the open fissure margins (arrow) in the basal lamina and to cells at the NR-RPE junction. mNtn1 was absent from this region in fused OFM seam at E12.5. (**f**) $Ntn1^{-/-}$ mice exhibited highly penetrant (~90%) bilateral coloboma (*arrows; n* = 10/11 homozygous E15.5-E16.5 animals analysed). (**g**) Cleft secondary palate (arrows) was observed in ~36% of $Ntn1^{-/-}$ embryos at E15.5-E16.5 (4/11 homozygous animals).

DOI: https://doi.org/10.7554/eLife.43877.016

The following figure supplements are available for figure 4:

**Figure supplement 1.** Developmental *NTN1* expression profiling in chick eye and OF.
DOI: https://doi.org/10.7554/eLife.43877.017

**Figure supplement 2.** Analyses of mouse *Ntn1* knockout fissures during fusion.
DOI: https://doi.org/10.7554/eLife.43877.018

**Figure supplement 3.** Gross ocular phenotype analyses of ntn1-deficient zebrafish.
DOI: https://doi.org/10.7554/eLife.43877.019

**Figure supplement 4.** Expression profiling for known interactors of NTN1.
DOI: https://doi.org/10.7554/eLife.43877.020

crossed heterozygote G0 fish ($ntn1a^{+/-}$) and observed several G1 embryos displaying bilateral ocular defects including coloboma and microphthalmia (***Figure 4—figure supplement 3***). DNA sequencing of the targeted *ntn1a* locus confirmed 100% (n = 3) of the phenotypic embryos were homozygous, whereas ocular defects or colobomas were not observed in any heterozygous (n = 6) or wild-type (n = 12) embryos. A recent study applied morpholino (MO) translation-blocking knockdown approaches to target *ntn1a* in zebrafish embryos and observed bilateral ocular colobomas in all fish injectected (***Richardson et al., 2019***), with normal early eye development and appropriately apposed fissure margins obvious prior to fusion. We were also able induce colobomas using MOs designed to target the translational start site of *ntn1a* (***Figure 4—figure supplement 3***). Bilateral colobomas were observed in 31/71 (43.7%) of MO injected embryos with no ocular phenotypes observed in control injections (n = 40). In combination, these results are in agreement with our data presented in chicken and mouse OFMs and that Netrin-1 is also essential for zebrafish eye development and is likely to have a specific role in tissue fusion. It also confirms an evolutionarily essential requirement for Netrin in ocular development, including OFC, in diverse vertebrate species.

## Discussion

### NTN1 is a strong candidate gene for coloboma and multisystem fusion defects

Our study provides strong evidence that Netrin-1 is essential for OFC in the developing vertebrate eye and is required for normal orofacial development and palate fusion. The transient and specific *NTN1* expression at the fusion plate, and the subsequent reduction/loss in fused OFM, suggests NTN1 has a direct role in the fusion process. Indeed, Netrin1-deficient mouse eyes displayed highly penetrant colobomas but their fissure margins were normally apposed during fusion initiation, arguing against a broad failure of early eye development. In further support for a direct role in epithelial fusion was previously published work showing fusion failure during development of the vestibular system of both chick and mice where *NTN1*-expression was manipulated (***Yung et al., 2015***; ***Salminen et al., 2000***; ***Nishitani et al., 2017***). In this developmental context, otic epithelia must fuse normally for the correct formation of the semicircular canal structures. Although we and others (***Richardson et al., 2019***) found coloboma in zebrafish knockdown experiments of *ntn1a*, we observed coloboma with microphthalmia in the context of complete knockout of *ntn1a*. This more severe phenotype in the complete absence of ntn1a implies there could be a more general requirement for Netrin-1 during early eye development, or could reflect teleost-specific eye developmental processes not shared among higher vertebrates (***Martinez-Morales et al., 2017***). Further work is

required to elucidate the precise role of Netrin-1 during OFC and broader eye development among different species.

Taken in combination, these findings strongly implicate NTN1 as a multipotent factor required for tissue fusion in multiple distinct developmental contexts. In humans, variants near *NTN1* have been associated with cleft lip in human genome wide association studies (*Leslie et al., 2016*; *Leslie et al., 2015*). While these are not monogenic disease mutations, this observation adds additional further relevance for future genetic studies of patients with coloboma. It is also consistent with our observations in *Netrin-1* knock-out animals having a high penetrance of both coloboma and cleft palate phenotypes. Therefore, we propose that *NTN1* should be included as a candidate gene in diagnostic sequencing of patients with human ocular coloboma, and should also be carefully considered for those with other congenital malformations involving defective fusion.

## NTN1 may have a role in CHARGE syndrome

Coloboma in association with additional fusion defects of the inner ear are two of the key clinical classifications for a diagnosis of CHARGE syndrome (*Verloes, 2005*). Further phenotypes commonly associated with the syndrome are septal heart defects and orofacial clefting, both with aetiologies likely to involve fusion defects (*Ray and Niswander, 2012*). CHARGE syndrome cases are predominantly caused by heterozygous loss-of-function pathogenic variants in the chromodomain helicase DNA-binding protein 7 (*CHD7*) gene (*Vissers et al., 2004*). Mice lacking Chd7 display CHARGE syndrome-like phenotypes and exhibit abnormal expression of *Ntn1* (*Hurd et al., 2007*; *Hurd et al., 2012*). In addition, ChIP-seq analyses have shown direct binding of Chd7 to the promoter region of *Ntn1* in mouse neural stem cells (*Engelen et al., 2011*). Given the amount of tissue available in the chick model, it would be possible and intriguing to confirm whether CHD7 directly regulates *NTN1* expression *in ovo* in the chick optic fissure. There is also emerging evidence that CHD7 and the vitamin A derivative retinoic acid (RA) indirectly interact at the genetic level during inner ear development (*Yao et al., 2018*). Defective RA signalling also leads to significant reduction of *Ntn1* expression in the zebrafish OFM (*Lupo et al., 2011*), implicating a possible genetic network involving RA and *CHD7*, where *NTN1* could directly mediate developmental fusion mechanisms from these hierarchical influences.

## How does Netrin-1 mediate fusion?

Netrin-1 is well-studied for its canonical roles in guidance of commissural and peripheral motor axons and growth-cone dynamics, with attraction or repulsion mediated depending on the co-expression of specific receptors (reviewed in *Lai Wing Sun et al., 2011*; *Larrieu-Lahargue et al., 2012*). We found that axonal processes were absent from the chick fissure margin during fusion stages, suggesting that the normal function of NTN1 may be to prevent axon ingression into the OFM to permit fusion. However, the phenotypic evidence from both the palate and vestibular system strongly support the argument that NTN1 has a non-guidance mechanistic role during OFC. Netrin orthologues have been recently associated with the regulation of cell migration and epithelial plasticity in the apparent absence of co-localised canonical Netrin-1 receptors (*Manhire-Heath et al., 2013*; *Lee et al., 2014*; *Yan et al., 2014*). In contrast, netrin acting together with its receptor neogenin combined to mediate close adhesion of cell layers in the developing terminal end buds during lung branching morphogenesis (*Srinivasan et al., 2003*). Although we observed strong *NTN1* expression in cells lining the chick OFM, and similar localisation of Netrin-1 protein in chick, human and mouse, we did not observe reciprocal expression of any canonical NTN1 receptors in our RNAseq datasets (e.g. UNC5, DCC or Neogenin; *Figure 4—figure supplement 4*). Indeed, the Netrin repulsive cue *UNC5B* was the most significantly downregulated DEG in fissure versus whole eye in our data and was also downregulated in human OFM (Sowden and Patel; *manuscript in preparation*). Therefore, it will be vitally important for future studies to elucidate interaction partners of Netrin in fusing tissues, or to reveal if Nerin-1 can act autonomously in these contexts and to provide deeper insight into its mechanistic function during fusion.

## The chick is a powerful model for OFC

The chick is one of the earliest established models for developmental biology and has provided many key insights into human developmental processes (*Stern, 2018*). Despite this, and extensive

historical study of eye development in chicken embryos, the process of chick OFC has not been well analysed until now. Indeed, the first study appeared only recently and specifically defined aspects of tissue fusion at the proximal (optic nerve and pecten) region of the OF (*Bernstein et al., 2018*), and did not observe complete fusion of epithelia in these regions. Indeed, closure of the proximal OF was characterised by intercalation of pecten and the lack of true epithelial continuum of neuroepithelial retina and RPE. By focusing on the epithelial fusion events in the distal and medial eye, our study complements the Bernstein et al study (*Bernstein et al., 2018*) to provide a comprehensive framework of OFC progression in the chick. Indeed, taken together, our analyses clearly define three distinct and separate anatomical regions in the developing chick OFM: the iris, the medial OFM, and the pecten. In addition, we present the spatial and temporal sequence of chick OFC at the anatomical and molecular level, and provide strict criteria for staging the process - based on a combination of broad embryonic anatomy, ocular, and fissure-specific features. Fusion initiated at the medial OFM at HH.St27/28 and continued until HH.St34, with predominantly distal to proximal directionality. In addition, we found that closure of the medial OFM is a true epithelial fusion process that occurs over a large time window of approximately 60 hr, involving two fusion plates, and that closes over 1.5 mm of complete fusion seam. This temporal window, the number of directly contributing cells, and the accurate staging of its progression allows unique opportunities for further experimentation. Importantly, one whole chick optic fissure (from HH.St29 onwards) can simultaneously provide data for unfused, fusing, and post-fused contexts.

In addition, our transcriptional profiling, including the identification of OFM-specific genes in the chick that include multiple human coloboma orthologues, builds on previous work that illustrate the chick as an excellent model for human eye development and the basis of embryonic malformations (*Wisely et al., 2017*; *Vergara and Canto-Soler, 2012*; *Trejo-Reveles et al., 2018*). These features, in combination with recent advances in chick transgenics and genetic manipulations (*Davey et al., 2018*), project the chick as a powerful to analyse cell behaviours during OFC and epithelial fusion. For example, the stable multi-fluorescent Cre-inducible lineage tracing line (the Chameleon chicken [*Davey et al., 2018*]) will be valuable to determine how the fissure-lining cells contribute to the fusing epithelia, while the very-recent development of introducing gene-targeted or gene-edited primordial germ cells into sterile hosts for germ-line transmission (*Taylor et al., 2017*) provides a rapid and cost-effective way to develop stable genetic lines to interrogate specific gene function (*Davey et al., 2018*; *Woodcock et al., 2017*). Thus, our study illustrates the powerful utility of the chick as a model for investigating OFC and for the discovery of novel candidate genes for coloboma, and is perfectly timed to coincide with major new developmental biology techniques in avian systems to place the chick model as a powerful addition to OFC and fusion research.

## Summary

This study provides the first detailed report of epithelial fusion during chick OFC and illustrates the power of the embryonic chick eye to investigate the mechanisms guiding this important developmental process further and to provide insights into human eye development and broader fusion contexts. We clearly define the temporal framework for OFC progression and reveal that fusion is characterised by loss of epithelial cell types and a coincidental increase in apoptosis. We reveal the specific expression of orthologues of known coloboma-associated genes during chick OFC, and provide a broad transcriptomic dataset that can be used to improve the identification of candidate genes from human patient exome and whole-genome DNA sequencing datasets. Finally, we identify that *NTN1* is specifically and dynamically expressed in the fusing vertebrate fissure - consistent with having a direct role in epithelial fusion, and is essential for OFC and palate development. We propose that *NTN1* should therefore now be considered as a new candidate for ocular coloboma and congenital malformations that feature defective epithelial tissue fusion.

## Materials and methods

**Key resources table**

*Continued on next page*

*Continued*

| Reagent type (species) or resource | Designation | Source or reference | Identifiers | Additional information |
|---|---|---|---|---|
| Reagent type (species) or resource | Designation | Source or reference | Identifiers | Additional information |
| Genetic reagent (*M. musculus*) | Ntn1-/- | PMID 26395479 | MGI:5888900 | Lisa Goodrich (Harvard Medical School, Boston MA). |
| Biological sample (*G. gallus*) | memGFP | PMID 25812521 | *Rozbicki et al., 2015* | Maintained at The Greenwood Building, Roslin Institute, UK. |
| Biological sample (*G. gallus*) | Chicken eye and OFM dissections | This paper | Hy-Line Brown | Maintained at The Greenwood Building, Roslin Institute, UK. |
| Antibody | NTN1 (Mouse monoclonal) | R and D Systems | MAB128 | one in 100 dilution for whole mount IF |
| Antibody | NTN1 (Rabbit polyclonal) | Abcam | ab126729 | one in 300 dilution for human and mouse IF; 1 in 500 dilution for chick cryosection IF |
| Antibody | Laminin-B1 (Mouse monoclonal) | DSHB | 3H11 | one in 20 dilution for all IF |
| Antibody | NF145 (Rabbit polyclonal) | Merk | AB1987 | one in 100 dilution for all IF |
| Antibody | Phospho-Histone H3A (Rabbit monoclonal) | Cell Signalling Technologies | #3377 | one in 200 for cryosections, 1 in 1000 for flat-mount |
| Antibody | Activated Caspase-3 (Rabbit polyclonal) | BD Pharminagen | #559565 | one in 400 dilution for all IF |
| Commercial assay or kit | Alexa Fluor Phalloidin (488 nm) | Thermo-Fisher | #A12379 | one in 40 dilution for all IF |
| Software, algorithm | Kallisto | PMID 27043002 | NA | NA |
| Software, algorithm | Limma | PMID 25605792 | NA | NA |

## Embryo processing

Hy-Line Eggs were incubated at 37°C at day 0 (E0), with embryo collection as stated throughout the text. Whole embryos were staged according to Hamburger Hamilton (*Hamburger and Hamilton, 1992*; *Hamburger and Hamilton, 1951*). Heads were removed and either ventral eye tissue was resected and flat-mounted and imaged immediately, or whole heads were placed in ice cold 4% paraformaldehyde (PFA) in pH 7.0 phosphate buffered saline (PBS), overnight and then rinsed twice in PBS. OFMs used for fusion progression measurements (flat mounts) were mounted in glycerol between a coverslip and glass slide, without fixation. Whole embryo, flat mounted OFMs, and dissected eye images for were captured on a Leica MZ8 light microscope and measurements were processed using FIJI (NCBI/NIH open source software [*Schindelin et al., 2012*]).

## Immunofluorescence

For cryosections, resected ventral chick eyes were equilibrated in 15% Sucrose-PBS then placed at 37°C in 7% gelatin:15% Sucrose, embedded and flash-frozen in isopentane at −80°C. Sections were cut at 20 μm. Immunofluorescence was performed on chick fissure sections as follows: 2 × 30 min rinse in PBS, followed by 2 hr blocking in 1% BSA (Sigma) in PBS with 0.1% Triton-X-100 [IF Buffer 1]. Sections were incubated overnight at 4°C with primary antibodies diluted in 0.1% BSA in PBS with 0.1% Triton-X-100 [IF Buffer 2]. Slides were then washed in 3 × 20 min PBS, followed by incubation

for 1 hr with secondary antibodies (Alexa Fluor conjugated with 488 nm or 594 nm fluorophores; 1:800–1000 dilution, Thermo Fisher), and mounted with ProLong Antifade Gold (Thermo Fisher) with DAPI. Alexa Fluor Phalloidin (488 nm; Thermo-Fisher #A12379) was added at the secondary antibody incubation stages (1:50 dilution). Human foetal eyes were obtained from the Joint Medical Research Council UK (grant # G0700089)/Wellcome Trust (grant # GR082557) Human Developmental Biology Resource (http://www.hdbr.org/). For Netrin-1 immunostaining in human and mouse tissues, cryosections were antigen retrieved using 10 mM Sodium Citrate Buffer, pH 6.0 and blocked in 10% Goat serum +0.2% Triton-X100 in PBS, then incubated overnight at 4°C with primary antibody (Abcam #ab126729; 1: 300) in block. Secondary antibody staining and subsequent processing were the same as for chick (above). For anti-NTN1 immunostaining in chick tissues, cryosections were hydrated in phosphate buffer (PB) pH7.2, antigen retrieved using 1% SDS in PB and blocked 2% bovine serum albumen +0.2% Tween-20 in PB (blocking buffer). Primary antibody was diluted in blocking buffer and incubated at room temperature for 4 days. Secondary antibody staining and subsequent processing were as stated above, but PB was used instead of PBS. For whole-mount immunofluorescence we followed the protocol from Ahnfelt-Rønne et al (*Ahnfelt-Rønne et al., 2007*), with the exception that we omitted the TNB stages and incubated instead with IF Buffer 1 (see above) overnight and then in IF Buffer two for subsequent antibody incubation stages, each for 24 hr at 4°C. No signal amplification was used. Antibodies were used against Phospho-Histone H3A and Netrin-1. Imaging was performed using a Leica DM-LB epifluorescence microscope, or a Nikon C1 inverted confocal microscope and Nikon EZ-C1 Elements (version 3.90 Gold) software. All downstream analysis was performed using FIJI. Image analysis for proliferation in the OFM on flat-mounts was performed by counting Phospho-Histone H3A positive foci using a region of interest grid with fixed dimensions of 200 $\mu m^2$ and throughout the entire confocal Z-stack. To quantitate apoptotic foci at the OFM, we used Activated-Casp3 immunofluorescence on serial cryosections of HH.St29-30 OFMs and collected confocal images for each section along the P-D axis. Image analysis was performed by counting A-Casp3 positive foci at the OFM in sequential sections using a region of interest with fixed dimensions of 100 $\mu m^2$. For histology and subsequent haematoxylin and eosin staining, resected eyes processed and image captured according to Trejo-Reveles et al (*Trejo-Reveles et al., 2018*).

## In situ hybridization

RNAscope was performed on HH.St29 cryosections using a probe designed specific to chicken *NTN1* according to Nishitani et al (*Nishitani et al., 2017*). For colourimetric in situ hybridisation, a ribprobe was for *NTN1* was designed using PCR primers to amplify a 500 bp product from cDNA prepared from chick whole embryos at HH.St28-32 (Oligonucleotide primers: Fwd 5'-ATTAACCC TCACTAAAGGCTGCAAGGAGGGCTTCTACC-3' and Rev 5'-TAATACGACTCACTATAGGCAC-CAGGCTGCTCTTGTCC-3'). The PCR products were purified and transcribed into DIG-labelled RNA using T7 polymerase (Sigma-Aldrich) and used for In Situ hybridization on cryosectioned chick fissure margin tissue (prepared as described above for immunofluorescence) or whole embryos using standard protocols (described in J. Rainger's doctoral thesis - available on request).

## Transgenic animal work

To obtain *Ntn1*[-/-] mouse embryos (Ntn1[tm1.1Good], RRID:MGI:5888900), we performed timed matings with male and female heterozygotes and took the appearance of a vaginal plug in the morning to indicate embryonic day (E)0.5. Embryos were collected at E11.5 and E16.6 and genotyped according to Yung et al (*Yung et al., 2015*). As with this previous report we observed ratios within the expected range for all three expected genotypes (28 total embryos: 13x *Ntn1*[+/-]; 10x *Ntn1*[-/-]; 5x WT – 46%; 35%; 18%, respectively). Embryos were fixed in 4% paraformaldehyde overnight and then rinsed in PBS and imaged using a Leica MZ8 light microscope. *Ntn1*[-/-] and C57Bl/6J animals were maintained on a standard 12 hr light-dark cycle. Mice received food and water ad lib and were provided with fresh bedding and nesting daily. For zebrafish work, we designed gene-editing sgRNA oligos alleles to target *ntn1a*: 5'-GGTCTGACGCGTCGCACGTG-3'. We then generated founder (G0) animals by zygotic microinjection of CRISPR/Cas9 components according to previous work (*Dutta et al., 2015*; *Varshney et al., 2015*; *Jao et al., 2013*). G0 animals were genotyped and used for crosses to generate G1 embryos which were scored for coloboma phenotypes and genotyped

individually (*Figure 4—figure supplement 3*). All experiments were conducted in agreement with the Animals (Scientific Procedures) Act 1986 and the Association for Research in Vision and Ophthalmology Statement for the Use of Animals in Ophthalmic and Vision Research (USA). Morpholinos were designed and generated by Gene Tools LLC (Oregon) to target the translation initiating site of *ntn1a*: 5′-CATCAGAGACTCTCAACATCCTCGC-3′, and a Universal control MO sequence was used as a control: 5′-ATCCAGGAGGCAGTTCGCTCATCTG-3′. One cell stage embryos were injected with 2.5 ng or 5.0 ng of ntn1a or control morpholino and allowed to develop to OFC stages (≥48 hpf). Oligos used for *ntn1a* genotyping by sanger sequencing were: 5′-TTACGACGAGAACGGACACC-3′ and 5′-GGAGGTAATTGTCCGACTGC-3′.

## Transcriptional profiling

For RNA seq analysis, we carefully dissected regions of (i) fissure-margin, (ii) ventral eye, and (iii) dorsal eye, and (iv) whole eye tissue from ≥10 individual embryos for each HH stage range (*Figure 3—figure supplement 1*). Samples were collected and pooled for each tissue type and stage to obtain n = 3 technical replicate RNA pools per tissue type per stage. Total RNA was extracted using Trizol (Thermo Scientific). Whole-transcriptome cDNA libraries were then prepared for each pool following initial mRNA enrichment using the Ion RNA-Seq Core Kit v2, Ion Xpress RNA-Seq Barcodes, and the Ion RNA-Seq Primer Set v2 (Thermo Scientific). cDNA quality was confirmed using an Agilent 2100 Bioanalyzer. Libraries were pooled, diluted, and templates were prepared for sequencing on the Ion Proton System using Ion PI chips (Thermo Scientific). Quantitative transcriptomics was performed using Kallisto psuedoalignment (*Bray et al., 2016*) to the Ensembl (release 89) chicken transcriptome. Kallisto transcript counts were imported into R using tximport (*Soneson et al., 2015*) and differentially expressed transcripts identified using Limma (*Ritchie et al., 2015*). Genes not expressed in at least three samples were excluded. To identify the relationships between samples, Log2 transformed counts per million were then calculated using edgeR (*Robinson et al., 2010*) and Spearman's rank correlation was used to identify the similarities in genome-wide expression levels between samples. All RNAseq data files are submitted to the NCBI Gene Expression Omnibus database (http://www.ncbi.nlm.nih.gov/geo) with the accession number GSE84916.

## Statistical analysis

Bar graphs display means ± SD or 95% confidence intervals as indicated. Sample sizes were n ≥ 3, unless stated otherwise. Statistical analyses were performed using Prism 8 (GraphPad Software Inc). Data were assessed for normal distribution by Shapiro-Wilk test where appropriate. Significance was evaluated by unpaired Student's t-test, where p≤0.05 was deemed significant. Asterisk indicate significance in *Figure 1* as *p≤0.05. **p≤0.01, ***p≤0.001.

## Acknowledgements

We wish to thank Megan Davey at Roslin Institute for academic discussions and support, David Fitz-Patrick at The MRC IGMM for supporting the RNAseq pilot experiments, Jenny Chen at NIE for technical assistance with zebrafish transgenics, Sadie Schlabach at HMS for help with embryo genotyping and sample collection, Richard Clark at The WTCRF in Edinburgh, and Agnes Gallagher at MRC IGMM for RNA-sequencing.

## Additional information

### Funding

| Funder | Grant reference number | Author |
|---|---|---|
| Fight for Sight UK | 1590/1591 | Joe Rainger |
| Company of Biologists | DMMTF-180520 | Joe Rainger |
| Biotechnology and Biological Sciences Research Council | BB/P013732/1 | Joe Rainger |
| Wellcome | ISSF3 | Joe Rainger |

| | | |
|---|---|---|
| Fight for Sight UK | Early Career Investigator Fellowship (1590/1591) | Joe Rainger |
| University of Edinburgh | Institutional Strategic Support Fund | Holly Hardy |
| Wellcome | Institutional Strategic Support Fund | Holly Hardy |
| Rosetrees Trust | | Aara Patel<br>Jane C Sowden |
| National Institute for Health Research Biomedical Research Centre at Great Ormond Street Hospital for Children and UCL | | Aara Patel<br>Jane C Sowden |
| Great Ormond Street Hospital Children's Charity | | Aara Patel<br>Jane C Sowden |
| Stuart HQ and Victoria Quan Fellow | | Andrea R Yung |
| Goldenson Faculty Research Grant | | Lisa V Goodrich |
| National Eye Institute | Intramural program | Sunit Dutta<br>Brian Brooks |

The funders had no role in study design, data collection and interpretation, or the decision to submit the work for publication.

### Author contributions
Holly Hardy, Formal analysis, Validation, Investigation, Methodology, Writing—review and editing; James GD Prendergast, Formal analysis, Methodology; Aara Patel, Sunit Dutta, Hannah Kroeger, Formal analysis, Investigation; Violeta Trejo-Reveles, Investigation, Methodology; Andrea R Yung, Resources, Visualization; Lisa V Goodrich, Resources, Supervision; Brian Brooks, Resources, Supervision, Validation, Writing—review and editing; Jane C Sowden, Resources, Formal analysis, Supervision, Validation, Writing—review and editing; Joe Rainger, Conceptualization, Data curation, Formal analysis, Supervision, Funding acquisition, Validation, Investigation, Visualization, Methodology, Writing—original draft, Project administration, Writing—review and editing

### Author ORCIDs
Holly Hardy ![ORCID] http://orcid.org/0000-0003-4603-7784
James GD Prendergast ![ORCID] http://orcid.org/0000-0001-8916-018X
Andrea R Yung ![ORCID] https://orcid.org/0000-0002-4053-378X
Joe Rainger ![ORCID] https://orcid.org/0000-0003-1091-5100

### Ethics
Human subjects: Human foetal eyes were obtained from the Joint Medical Research Council UK (grant # G0700089)/Wellcome Trust (grant # GR082557) Human Developmental Biology Resource (http://www.hdbr.org/). The consent, use and disposal of HDBR samples is regulated by the UK Human Tissue Authority (HTA). The HDBR is a Research Ethics Committee (REC) approved and HTA licenced tissue bank. This means that most research projects based within the UK do not need to obtain their own REC approval.

Animal experimentation: All animal work was carried out in strict accordance with the United Kingdom Home Office Animal (Scientific Procedures) Act 1986. All chicken experiments, breeding and care procedures were approved and carried out under license from the UK Home Office (PPL 7008940 - Prof Helen Sang) and subject to local ethical review by the Roslin Institute AWERB. No regulated procedures were used in this study. Generation and maintenance of memGFP flock were in accordance with annex III of Directive 2010/63 EU and Home Office Codes of Practice. All mouse and zebrafish work was conducted in compliance with protocols approved by the Institutional Animal Care and Use Committee at Harvard Medical School, and at The NIH National Eye Institute. Mice were used from an existing study (Yung et al., Development. 2015). Ntn -/- (Ntn1tm1.1Good,

MGI:5888900) and C57Bl/6J animals were maintained on a standard 12hr light-dark cycle. Mice received food and water ad lib and were provided with fresh bedding and nesting daily. All experiments were conducted in agreement with the Animals (Scientific Procedures) Act 1986 and the Association for Research in Vision and Ophthalmology Statement for the Use of Animals in Ophthalmic and Vision Research. Pregnant dams were anaesthetised by $CO_2$ asphyxiation and euthanised by cervical dislocation. Embryos were collected at E11.5, E15.5 and E16.5. All embryos were immediately culled on ice by decapitation. All zebrafish embryos/larvae are taken at between 30 hpf-56 hpf and immediately anaesthetised with tricaine methane sulfonate (MS222, 168 mg/l) on ice. Embryos are then euthanised in bleach solution (sodium hypochlorite 6.15%) in water at 1 part bleach to 5 parts water. The larvae remain in this solution at least five minutes prior to disposal to ensure death.

### Decision letter and Author response
Decision letter https://doi.org/10.7554/eLife.43877.025
Author response https://doi.org/10.7554/eLife.43877.026

## Additional files

### Supplementary files
• Transparent reporting form
DOI: https://doi.org/10.7554/eLife.43877.021

### Data availability
All RNAseq data files are submitted to the NCBI Gene Expression Ominibus database (http://www.ncbi.nlm.nih.gov/geo) with the accession number GSE84916.

The following dataset was generated:

| Author(s) | Year | Dataset title | Dataset URL | Database and Identifier |
|---|---|---|---|---|
| Rainger J | 2019 | Segmental chick eye transcriptome analysis | http://www.ncbi.nlm.nih.gov/geo/query/acc.cgi?acc=GSE84916 | NCBI Gene Expression Omnibus, GSE84916 |

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
