## [Decision Letter]

Thank you for submitting your article "Detailed analysis of chick optic fissure closure reveals Netrin-1 as a conserved mediator of epithelial fusion" for consideration by *eLife*. Your article has been reviewed by Marianne Bronner as the Senior Editor, a Reviewing Editor, and three reviewers. The following individuals involved in review of your submission have agreed to reveal their identity: Stephan Heermann (Reviewer #2); Teri Belecky-Adams (Reviewer #3).

I am including the three reviews at the end of this letter, as there are a variety of specific and useful suggestions in them. Some of the comments have to do with clarity of presentation, critical interpretation of the data, and comparison with previously published data – these comments should be relatively straightforward to address. The more substantive ones relate to the experimental evidence that Netrin plays a causal role in fissure closure.

We appreciate that the reviewers' comments cover a broad range of suggestions for improving the manuscript. Please use your best judgment in deciding which of these can be accommodated in a reasonable period of time. We look forward to receiving your revised manuscript.

*Reviewer #1:*

This manuscript addresses the question of how epithelial fusion occurs within a specific structure in the eye, the optic/choroid fissure. Epithelial fusion is not well understood, and is a feature in the development of a number of different organ systems. The authors describe events that occur during optic fissure fusion in chick embryos, then carry out transcriptome analysis on microdissected tissues in an effort to identify factors that might be directly involved in the fusion process. They identify NTN1 as a factor enriched in the fissure, and then move to mouse and zebrafish for loss-of-function phenotypes.

The work to describe optic fissure fusion in chick is fundamentally sound, although more schematics are necessary to orient the reader (especially a broader audience), and editing should be done to more explicitly state how this work is different (more focused, finer time window; more quantitative analysis) from a recent publication of describing many aspects of chick optic fissure development (Bernstein et al., 2018). The transcriptome analyses are excellent, and the temporal comparisons are useful.

On the other hand, mechanistic conclusions are weaker and often correlative. For example, based on antibody staining at a single timepoint, roles for proliferation and apoptosis are inferred or suggested; conclusions would be much stronger if proliferation and apoptosis had been tested via inhibition (for example, with pharmacological reagents). The idea of netrin as an epithelial fusion factor is exciting and intriguing, but the results are difficult to interpret in this form. The netrin mouse mutant exhibits coloboma, but given that the authors review many direct and indirect causes of coloboma in the introduction, it is not clear that netrin acts in mouse in the way they hypothesize from chick data. Characterization of the mutant phenotype (along with controls) would be necessary to interpret the gross morphological phenotype. The zebrafish data are also difficult to interpret, and no molecular characterization of these novel mutant alleles is shown. Similarly, characterization and controls in zebrafish would be necessary to interpret these data.

Abstract: "Our data reveal that NTN1 is a new locus for human coloboma…"

Unless there are human genetic data, this statement does not seem to be supported by the manuscript in this form.

More information is needed to understand the images (especially the fluorescent confocal images) in the manuscript. For example, in Figure 1C, is this a single confocal optical section? Or a projection? This is important especially for interpreting data such as in Figure 4 (netrin localization; see below).

Table 1 and Table 2: The authors have carried out quantitative analysis of optic fissure fusion in chick. Yet these tables are difficult to interpret, which makes this section of the results confusing. For example, subsection “OFC in the chick occurred within a wide spatial and temporal window”: "FP2 displayed active movement in a posterior direction to create a fused seam between FP1-FP2 that extended until HH.St34 (Table 2)". FP1 and FP2 are not noted in the table, and in the table, the term "fused collar" is confusing. Inclusion of a schematic or a graphical representation of these data (including where these terms fit with respect to the measurements being taken) might help. In addition, the authors note four distinct phases of fusion, with #3 being "active fusion as two FPs separate to generate a fused seam along the A-P axis" (subsection “OFC in the chick occurred within a wide spatial and temporal window”). Can the authors comment on the variability of this in Table 1? The text suggests that there should be two FPs at these stages, but at all of these stages, 50% of the embryos have 1 FP.

Figure 1 and Figure 2 (and associated supplemental figures): regarding the proliferation and apoptosis analysis, the antibody staining is clear at the stage shown. But this is correlative and would not seem to be sufficient to exclude or support a role for either process in optic fissure fusion: can the authors use pharmacological reagents to inhibit proliferation or apoptosis to functionally test their roles?

Figure 2—figure supplement 1: regarding RGC axons, in this orientation, it is difficult to interpret the localization of NF-145 staining. RGC axons, and their positions relative to the fusing fissure, might be easier to see in a fissure flatmount. This may be important, since the authors use these data to exclude a role for RGC axons (and possibly a role for netrin acting through RGC axon guidance) in optic fissure fusion.

Figure 3, also subsection “Transcriptional profiling reveals genetic conservation between chick and human OFC”: the figure suggests that St28-32 was used to represent "Active-fusion", but the results state Hh.st28-30. Please clarify.

Figure 4B’’: the authors state that netrin protein is "specifically enriched in open fissures but not in the fused seam". Although netrin protein may be missing from the specific site where fusion took place, it does still appear to be present in the basal lamina (although it seems absent in Figure 4B – was this a single confocal optical section, and is the remaining netrin "inside" after fusion?). Do the authors think that netrin may be carrying out a different function around the optic nerve after fusion? How does this fit with their model?

Figure 4, regarding netrin expression: more expression data is required, earlier in development. At this stage, netrin RNA is absent from the fused seam, but was it actually present earlier in this region's pre-fusion margins? Might it mark the first site of fusion? A fissure flatmount could be helpful for assessing expression.

Figure 4, regarding mouse mutant data: these data are intriguing, but not enough is shown to demonstrate that netrin is mediating fusion. What is the more detailed phenotype of the mutant mouse? Does loss of netrin affect localization and movement of periocular mesenchyme or RGC axons? Coloboma could be caused via indirect effects through many other factors.

Figure 4—figure supplement 1: the zebrafish data are difficult to interpret, as no molecular characterization of these novel mutant alleles is included. Can the authors demonstrate that these are loss-of-function alleles (for example, via other phenotypes)? Is mRNA still present? In addition, the phenotypic penetrance seems unclear. In subsection “Loss of Netrin causes coloboma and multisystem fusion defects in vertebrates”: "We observed bilateral ocular coloboma in all homozygous mutant animal(s) analysed…"; whereas in subsection “Immunofluorescence”: "In these crosses, we observed coloboma phenotypes in 4/28 offspring, giving approximately 50% penetrance…" Were zebrafish mutant embryos genotyped? This is not clear, and embryos should be genotyped. Other control experiments and further characterization of the mutant phenotype should be carried out (e.g. potential effects periocular mesenchyme migration or RGC axons) in order for the role of netrin as a fusion mediator to be more convincing.

Regarding both mouse and zebrafish mutant data: the move to the other organisms leads to basic questions about optic fissure fusion and netrin expression in those other organisms – are the same fusion mechanisms at play at a descriptive level? Is netrin really expressed along the entire optic fissure margins in mouse and zebrafish at the appropriate timepoints? More characterization of these other systems would be necessary to interpret these phenotypes.

*Reviewer #2:*

In the manuscript "Detailed analysis of chick optic fissure closure reveals Netrin-1 as a conserved mediator of epithelial fusion", Hardy and colleagues carefully addressed the fusion process of the medial and distal optic fissure domain in chicken embryos and by analysis of fixed samples at different developmental stages identified the onset and the progression of fusion. They also show that cell proliferation was not increased in the fissure margins compared to other regions but that apoptosis could be found in and next to the progressing fusion site. Furthermore, they applied transcriptomic analysis in order to identify genes, which are important during fissure fusion. Among others they identified Netrin-1 (NTN1), which was found most specifically upregulated during fusion in chicken. NTN1 expression and protein localization was found in the fissure region mainly before and during fusion. Protein localization was then also checked in developing human- and mouse eyes. And the loss of NTN1 was addressed in mouse and zebrafish.

Per se this is an important contribution to the field. It is beneficial for the field to include also the chicken model for studying optic fissure fusion. In itself the data seem sound. However, the fusion of the fissure in chicken was addressed also recently (Bernstein et al., 2018). Notably, the data (e.g. onset of apposition, BM modulation and fusion) presented there was different to the presented work here. Could there be strain dependency? Hardy et al. state in the discussion that Bernstein et al. did not report fusion in anterior or midline region and mentioned that in the proximal regions fusion is happening via intercalation. Nevertheless, Bernstein et al. reported fusion. It should be pointed out which of the current findings is new, which is backing up previous data and which is potentially at odds. Along the lines of chicken as a model for fissure fusion, it would be helpful to address the role of the pecten. But beyond that, Hardy et al. provide transcriptional data from well-designed experiments and further addressed the necessity of NTN1.

A lot of effort was put into the analysis of loss of NTN1 in other species, mouse and zebrafish. Technically it would be possible to address the loss of NTN1 also in chicken. What was the rational to use the other species only?

It was proposed that the role of NTN1 would be the same in the different species. However, in zebrafish the remaining fissure seems a bit wider. What is the age of the embryo? It would be helpful to see also histological images to see that the margins were in "apposition" properly.

Is the work with human samples covered by the HDBR? In the ethics statement it is mentioned that in the UK most studies will not need specific approval, but is this study also covered?

In some cases, I was a bit lost in the images presented e.g. Figure 1C. How was the staining achieved? What exactly do we see? Are the cells in FP1 head to head and in FP2 side by side? What is the scale and orientation of the individual images? I will give more examples in the minor point section.

It would be good if the data mentioned in subsection “Chick OFC was characterised by the breakdown of basement membranes, loss of epithelial morphology and localised apoptosis” and subsection “*Netrin-1* is specifically and dynamically expressed in the fusing OFM” could be shown.

In some cases, the citations seemed incorrect, e.g. subsection “Transcriptional profiling reveals genetic conservation between chick and human OFC”, Cho and Cepko (2006).

Subsection “Transcriptional profiling reveals genetic conservation between chick and human OFC”: what would be the definition of "biologically significant" in terms of differential gene expression? Would this not be dependent on the gene and gene product? Can this be generalized?

Subsection “Transcriptional profiling reveals genetic conservation between chick and human OFC”: Table 2 is not showing this. Figure 3B shows the 12 increased only.

Figure 1A: Is the OFM really the unpigmented region? Could it be the fissure itself?

Figure 1B: To what extent is the iris developed at the stage of fissure fusion? The pecten could be explained more. On the images, it appears as a gap. Could the axes be renamed to proximal and distal?

Figure 1C: What is the orientation, magnification and how was the labelling achieved?

Figure 1E: Were this and Figure 1—figure supplement 1G independent datasets?

Figure 1F: Is the proliferation increased outside or reduced inside?

Figure 1—figure supplement 1B: What is seen here?

Figure 1—figure supplement 1C: Is this taken from flat mounted tissue? What is the orientation?

Figure 1—figure supplement 1D: It is stated that at HH26 there is no fusion plate. Could higher magnifications be provided? In Figure 1—figure supplement 1Di it is not totally clear, especially comparing it to Figure 2B.

Figure 1—figure supplement 1G: The label "G" is missing in the figure. I do not understand the note that at HH29 the seam length was too small to quantify. The length itself was quantified in Figure 1—figure supplement 1F.

Table 1: Can an animal have FP1 and FP2? At stage HH29 n 5 were analyzed. 4 showed 2FP 5 showed 1FP. Could this be explained?

Figure 2:

Without time-lapse analysis, the term dynamics is potentially misleading.

Figure 2A: Arrowheads in fusion plate image point to a domain in which the basal lamina is still visible.

Figure 2B: Can the authors be sure that the future NR and RPE are both contributing and are both remodeled? Could there be dynamic in the fissure margins?

Figure 2C: The image presented to show the apposition is not corresponding to Figure 2B.

Figure 2D: Which domain was used for quantification? What was considered seam and outside seam and what could be considered OFM and outside OFM (towards nasal and temporal directions respectively)?

Figure 2—figure supplement 1: What is meant by medium? What is the orientation? What is considered central retina?

Figure 3B: Could the boxes be separated for the different stages? The annotation over the boxed could be misunderstood for statistical data. The annotation (.) should be included in the legend. The legends mention CHDL1 and not CHRDL1.

Figure 3—figure supplement 1A: The images are appreciated. Were the individual regions dissected manually and in independent eyes also for fissure and ventral eye, meaning, is ventral eye including the fissure?

Figure 3—figure supplement 1B: How was the dorsal data acquired? Is the transcriptional data available?

Figure 4A-C: What is the orientation?

Figure 4B: What is meant by whole mount? At the FP2, the NTN1 signal seems less intense. Is the arrow positioned correctly? A co-staining with a basal lamina marker would be helpful to conclude the localization.

Figure 4B’’: What is NT? Is the overall annotation correct? Is NTN1 expressed also in the POM?

Figure 4C, D: Higher magnifications are needed and potentially a co-labelling with a basal lamina marker.

Figure 4F: What is shown in the second image?

Figure 4—figure supplement 1B: The anti-laminin staining looks odd.

Figure 4—figure supplement 1C: What was the age of the embryos? Arrows were used twice for different annotations. In the no antibody control, green signal can be seen. Was there a problem with color channel separation? It was stated that there was no staining.

Figure 4—figure supplement 1D: Please add axes for orientation and landmarks for orientation.

Figure 4—figure supplement 1F: If the penetrance of a phenotype is given, could it be also supported by genotyping besides by calculation? Was the phenotype consistent with the image in f? The margins seem farther apart? Where the margins in "apposition" correctly? Histology should be performed, or confocal analysis, to address this issue. What is the age of the embryo presented?

At the end of the legend there is a redundant text section.

It would be good if the data mentioned in subsection “Chick OFC was characterised by the breakdown of basement membranes, loss of epithelial morphology and localised apoptosis” and subsection “*Netrin-1* is specifically and dynamically expressed in the fusing OFM” could be also shown.

The RNAseq data was shared. It would be best if it was accessible also without too much bioinformatics background.

*Reviewer #3*

Summary: The authors have (1) characterized in detail the fusion of the coloboma in chick embryos, (2) identified sets of genes that change across several stages and in relation to other areas of the developing chick optic cup (dorsal, ventral, and whole optic cup), and (3) tested the role of netrin in chick, mouse, and zebrafish optic fissure closure. The reviewer congratulates the authors on a beautifully done study.

---

## [Author Response]

Reviewer #1:[…]Abstract: "Our data reveal that NTN1 is a new locus for human coloboma…"Unless there are human genetic data, this statement does not seem to be supported by the manuscript in this form.

We have made this statement more accurately reflect the findings of the study. This section of the Abstract now reads: “Our data suggest that NTN1 is a strong candidate locus for human coloboma and other multi-system developmental fusion defects, and show that chick OFC is a powerful model for epithelial fusion research.”

More information is needed to understand the images (especially the fluorescent confocal images) in the manuscript. For example, in Figure 1C, is this a single confocal optical section? Or a projection? This is important especially for interpreting data such as in Figure 4 (netrin localization; see below).

We apologise for not making this more clear in the figures or legends and have improved the information available in the manuscript to help with interpretation of the data and figure legends have been amended to clearly describe the data shown.

Table 1 and Table 2: The authors have carried out quantitative analysis of optic fissure fusion in chick. Yet these tables are difficult to interpret, which makes this section of the results confusing. For example, subsection “OFC in the chick occurred within a wide spatial and temporal window”: "FP2 displayed active movement in a posterior direction to create a fused seam between FP1-FP2 that extended until HH.St34 (Table 2)". FP1 and FP2 are not noted in the table, and in the table, the term "fused collar" is confusing. Inclusion of a schematic or a graphical representation of these data (including where these terms fit with respect to the measurements being taken) might help.

We appreciate the reviewer’s efforts to interpret our data as presented and agree that the way we had presented this was confusing. We have made the following amendments to improve the interpretation and clarity of the measurement data: (i) we have included a schematic in Figure 1 to orientate the reader, which includes labelled anatomical features and axes for orientation; (ii) we have added more samples to our data and stratified the data in Table 1 to more clearly indicate the presence of either a single fusion point or two fusion points; (iii) we have removed the data for the “fused collar” measurements in Table 2 as this does not add any useful additional information to the table or study; (iv) we have also significantly edited the text in the results paragraph relating to this data to improve clarity and interpretation. This now reads:

“Fusion was first initiated between HH.St27-28 as confirmed by the definitive appearance of joined epithelial margins at a single fusion point (FP). […] The process is active between HH.St27-HH.St34 and proceeds over ~66 hours.”

In addition, the authors note four distinct phases of fusion, with #3 being "active fusion as two FPs separate to generate a fused seam along the A-P axis" (subsection “OFC in the chick occurred within a wide spatial and temporal window”). Can the authors comment on the variability of this in Table 1? The text suggests that there should be two FPs at these stages, but at all of these stages, 50% of the embryos have 1 FP.

The reviewer raises an important point as we observed subtle variability in the progression of fusion within defined HH stages of development, despite using meticulous adherence to established staging criteria using developmental landmarks. Indeed, this variation should be taken into account when analysing chick OFC processes, as is the case with OFC in other model organisms. However, our model attempts to be inclusive of these variabilities and remains a useful staging framework for studies of chick OFC. For example, in our model “initiation” occurs between HH.St27-28 where 5/7 OFMs (>70%) analysed had observable fusion plates. However, only one OFM (at HH.St28) had an expanded seam with two clear FPs, whereas the majority had only one observable FP. Two had no FPs and no FPs were identified prior to these stages. Subsequently, “active fusion” occurs from HH.St28-33. Our data now shows that in this range 19/24 OFMs (79%) analysed have 2x FPs (i.e. both FP1 and FP2 present). In contrast, by HHSt.34 (“complete fusion”) we could not definitively identify any proximally open fissures in all samples analysed, indicating that the seam had met the pecten and active fusion had been completed (in 100% of samples).

Figure 1 and Figure 2 (and associated supplemental figures): regarding the proliferation and apoptosis analysis, the antibody staining is clear at the stage shown. But this is correlative and would not seem to be sufficient to exclude or support a role for either process in optic fissure fusion: can the authors use pharmacological reagents to inhibit proliferation or apoptosis to functionally test their roles?

We were careful not to assign a functional role for either of these processes but believe both observations are important to accompany the fusion-progression data to establish the chick model of OFC. We specifically included the apoptosis data because the role and requirement for apoptosis during epithelial fusion has been controversial in other animal models of OFC and fusion. Whether apoptosis is indeed necessary or has a direct role in mediating fusion is still undetermined and may vary across divergent species, but our data shows strong spatial and temporal correlation between apoptosis and fusion in the chick OFC. We are currently developing tools to inhibit apoptosis at both the chemical and genetic level in chick, and to coincidentally manipulate Netrin-1 expression and function in this context. We feel these studies are beyond the scope of the current study but will provide valuable information of the specific pathways and processes that lead to cell-death in the chick fissure.

On reflection of the reviewer’s comment, we have moved the proliferation data into supporting data as although the analysis clearly showed fewer dividing cells in the fused seam, we feel that this observation, while interesting, requires additional analyses to reveal the exact mechanisms of seam expansion. Inhibition of cell-proliferation, as suggested by the reviewer, would need to specifically and reproducibly target cells within the OFM to formally rule out a requirement for proliferation in seam expansion. However, these tools are not yet available.

Figure 2—figure supplement 1: regarding RGC axons, in this orientation, it is difficult to interpret the localization of NF-145 staining. RGC axons, and their positions relative to the fusing fissure, might be easier to see in a fissure flatmount. This may be important, since the authors use these data to exclude a role for RGC axons (and possibly a role for netrin acting through RGC axon guidance) in optic fissure fusion.

We agree with the reviewer and have now included flat mounted brightfield OFM image data to help with orientation in this figure. We have also provided additional flat-mount immunofluorescence analyses, which confirm the absence of NF145 in the fusing and nascently fused OFM, supporting our previous conclusion that RGC axons do not have a direct role in mediating chick OFC.

Figure 3, also subsection “Transcriptional profiling reveals genetic conservation between chick and human OFC”: the figure suggests that St28-32 was used to represent "Active-fusion", but the results state Hh.st28-30. Please clarify.

We are grateful for this observation. The segmental dissections for RNAseq were performed at HHSt.28-30, during the active fusion period. We did not dissect at any later stages during active fusion. We have amended these inconsistencies throughout the manuscript and figure.

Figure 4B’’: the authors state that netrin protein is "specifically enriched in open fissures but not in the fused seam". Although netrin protein may be missing from the specific site where fusion took place, it does still appear to be present in the basal lamina (although it seems absent in Figure 4B – was this a single confocal optical section, and is the remaining netrin "inside" after fusion?). Do the authors think that netrin may be carrying out a different function around the optic nerve after fusion? How does this fit with their model?

The dynamic nature of NTN1 is intriguing. The images the reviewer refers to are indeed single-plane confocal sections and this has been made clearer in the figure legend. The data shows the absence of NTN1 within the fused seam and is consistent with NTN1 mRNA localisation. In our mouse data (Figure 4—figure supplement 2), we observed continual Ntn1 localisation in the optic nerve after fusion was completed. Although we have not analysed NTN1 localisation in the chick optic nerve at equivalent stages, we believe that in both species Netrin-1 has a role in the continued development of the optic nerve that progresses after fusion has completed in the medial OFM, consistent with a role in axonal guidance or neuronal migration. The exact relevance of our localisation data in the OFM is unclear but suggests there may be further multiple roles for NTN1 during the progression of fusion that are mediated by its differential localisation in the basal lamina and ECM. We are actively working on elucidating these roles using in vitro and our chick OFC model.

Figure 4, regarding netrin expression: more expression data is required, earlier in development. At this stage, netrin RNA is absent from the fused seam, but was it actually present earlier in this region's pre-fusion margins? Might it mark the firstsite of fusion? A fissure flatmount could be helpful for assessing expression.

We have added whole mount in situ hybridisation data showing NTN1 expression in the ventral optic cup before the OFM is established (HH.St22-24; Figure 4—figure supplement 1). Our transcriptomic analysis had revealed NTN1 as the most highly-expressed fissure-specific gene in our dataset, including at pre-fusion stages (HH.St25-27). We also showed in situ data of NTN1 expression in cryosections at the open medial OFM immediately prior to fusion (Figure 4—figure supplement1A), confirming its localisation in the OFM leading up to fusion. We have now added immunofluorescence data on cryosections confirming that NTN1 protein is also localised to this region immediately preceding fusion. In combination, these data show that NTN1 is expressed throughout the OFM long before fusion is initiated, and therefore is not specific to the fusion initiation point. This suggests that NTN1 has a role in mediating fusion at all points along the A-P axis and is important for establishing or maintaining the OFM in the developing ventral eye.

Figure 4, regarding mouse mutant data: these data are intriguing, but not enough is shown to demonstrate that netrin is mediating fusion. What is the more detailed phenotype of the mutant mouse? Does loss of netrin affect localization and movement of periocular mesenchyme or RGC axons? Coloboma could be caused via indirect effects through many other factors.

We were careful not to assign a specific mechanism for NTN1-mediated fusion in our manuscript as we are firmly aware that this will require significant additional work to elucidate. However, a requirement for Netrin-1 in the optic fissure was shown in zebrafish in a report published while our manuscript was under review (Richardson et al., 2019). In their study, transient morpholino gene knockdown experiments of the zebrafish orthologue *ntn1a* resulted in a coloboma phenotype. The authors provided some additional phenotypic characterisation that showed correct apposition of the fissure margins in *ntn1* deficient embryos. Consistent with this, and in further support of a direct role for Netrin-1 in the fusion process, we have added additional phenotypic data from sections of Ntn1^-/-^ eyes at stages when fusion initiates in the mouse eye (E11.5; Figure 4—figure supplement 2). This data showed that early growth of the eye in Ntn1^-/-^ embryos was sufficiently normal to bring the opposing optic fissure margins in direct apposition. Furthermore, the tissue morphology in the OFM appeared indistinguishable from the wild type controls and there was no evidence of ectopic cells or axonal processes between the apposed OFM edges. We believe that these new data, in combination with (i) the highly-specific expression of Netrin-1 in multiple vertebrates species during fusion in our study, (ii) prior observations of fusion-specific defects in the vestibular system resulting from Netrin-1 mis-regulation, and (iii) our additional findings of palate fusion defects in the Ntn1^-/-^ mice, all support a fusion-specific role for Netrin-1.

Figure 4—figure supplement 1: the zebrafish data are difficult to interpret, as no molecular characterization of these novel mutant alleles is included. Can the authors demonstrate that these are loss-of-function alleles (for example, via other phenotypes)? Is mRNA still present? In addition, the phenotypic penetrance seems unclear. In subsection “Loss of Netrin causes coloboma and multisystem fusion defects in vertebrates”: "We observed bilateral ocular coloboma in all homozygous mutant animal(s) analysed…"; whereas in subsection “Immunofluorescence”: "In these crosses, we observed coloboma phenotypes in 4/28 offspring, giving approximately 50% penetrance…" Were zebrafish mutant embryos genotyped? This is not clear, and embryos should be genotyped. Other control experiments and further characterization of the mutant phenotype should be carried out (e.g. potential effects periocular mesenchyme migration or RGC axons) in order for the role of netrin as a fusion mediator to be more convincing.

The work presented in our first submission was performed on a gene edited (GE) line produced as part of a wider study, and both the *ntn1a* and *ntn1b* loci were targeted. This line therefore required a complex crossing and genotyping strategy to establish the causation of the colobomas observed (4/28) of all offspring. We had observed no colobomas in F1 crosses on the *ntn1b* mutant background and the colobomas were only present when the *ntn1a* allele was introduced to the cross. However, we were unable to maintain this line for subsequent analyses as the long-term viability and fertility of *ntn1a^+/-^:ntn1b^-/-^* animals was very poor.

Therefore, we have now derived an additional line with only the *ntn1a* locus edited. Using the same sgRNA GE design we generated a frame-shift nonsense mutation (p.Cys90Ala.fs15; see Figure 4—figure supplement 3), and the data presented in the revised manuscript is solely for this line, with all G0, F0 and F1 animals individually genotyped (either *ntn1a*^+/-^, *ntn1a*^-/-^, or Wt). We also have applied morpholino gene-knockdown of *ntn1a* in zebrafish embryos. These experiments support the *ntn1a* GE data, that loss of *ntn1a* causes highly penetrant colobomas.

While our work was under review, a paper from the group of Mariya Moosajee (Richardson et al., 2019) described transcriptional profiling analyses in the zebrafish ventral optic cup during OFC. Remarkably, they revealed specific upregulation of *ntn1a* coincident with the progression of fusion, with subsequent down-regulation as fusion progressed, similar to our observations in chick. They also provided morpholino knockdown experiments of zebrafish *ntn1a* with an ocular coloboma phenotype. Consistent with our data from mouse and zebrafish, the mutant phenotype was associated with normal early growth of the ventral optic cup and the tips of the optic fissure margins coming into close contact with each-other. No intervening POM or axonal processes were observed. Therefore, the Richardson et al. study adds further evidence of a fusion-specific role for Netrin-1 in OFC.

Regarding both mouse and zebrafish mutant data: the move to the other organisms leads to basic questions about optic fissure fusion and netrin expression in those other organisms – are the same fusion mechanisms at play at a descriptive level? Is netrin really expressed along the entire optic fissure margins in mouse and zebrafish at the appropriate timepoints? More characterization of these other systems would be necessary to interpret these phenotypes.

The reviewer raises an important question of the evolutionary conservation of fusion mechanisms and we agree with the questions that are raised. Our paper exists as a unique study of fusion at the tissue and genetic level and supports the chick as a solid model for ongoing fusion-specific research. Many additional studies are now emerging in other model organisms applying genetic and imaging data to elucidate fusion mechanisms that reveal both commonality and differences between divergent species. The recent study by Richardson et al. (2019) performed transcriptional profiling in the zebrafish OFM and showed *ntn1a* is expressed specifically during fusion stages. Our data in mouse also show this at the protein level, and both are consistent with the specific NTN1 expression we observed in the chick OFM. In combination with our phenotypic data, we believe these strongly support the hypothesis of a fusion-specific role for Netrin-1 that is common to all vertebrates. Further work is required to elucidate these mechanisms and address the points the reviewer raises – indeed these form the basis for the long-term research focus of our lab.

Reviewer #2[…]A lot of effort was put into the analysis of loss of NTN1 in other species, mouse and zebrafish. Technically it would be possible to address the loss of NTN1 also in chicken. What was the rational to use the other species only?

While overexpression and morpholino analyses are indeed possible in the chick embryo, loss of function studies are more difficult and not well established in the literature. In particular, studies of genes that are secreted (such as Netrin-1) or are predicted to induce cell non-autonomous phenotypes will require highly consistent and widespread cellular transduction of plasmids or RNAi payloads. Current efforts are specifically focused on optimising such approaches and the development of germline knock-out chicken lines, however, these approaches require significant optimisation and costs, respectively and are not as yet “online”. For the purpose of this paper, we chose the mouse and zebrafish knock-out models as they were readily available. However, the data from these models strongly suggests that the requirement for Netrin-1 is highly conserved among evolutionarily diverse vertebrates and our transcriptomic data is consistent with many other studies recently becoming available in these model organisms. Whether the functional mechanisms are consistent among these species remains to be proven but the localisation and then removal from netrin specifically in epithelial cells at the fissure margin in humans, mice and chick are intriguing and support a common mechanism.

It was proposed that the role of NTN1 would be the same in the different species. However, in zebrafish the remaining fissure seems a bit wider. What is the age of the embryo? It would be helpful to see also histological images to see that the margins were in "apposition" properly.

We have added new data to show coloboma phenotypes in both morpholino knockdown and gene edited knockout embryos, at 48hfp, when apposition occurs and fusion initiates. This new data is intriguing in the apparent phenotypic difference between zebrafish and mouse in the absence of Netrin-1. We are keen to understand the mechanisms that lead to coloboma in both zebrafish and mice lacking Netrin-1 but such work is out with the scope of this current study. We hope to provide robust phenotypic analyses in such future studies.

Is the work with human samples covered by the HDBR? In the ethics statement it is mentioned that in the UK most studies will not need specific approval, but is this study also covered?

The HBDR has ethical approval to collect and distribute embryonic and fetal material to all UK based research projects, without the need for individual researchers to obtain their own project specific ethics, providing their project fits within the remit of the HDBR approval. This study is covered for the use of human samples and is in accordance with the HBDR (Newcastle Ethics Form Approval Terms and Conditions – Section 4).

In some cases, I was a bit lost in the images presented e.g. Figure 1c. How was the staining achieved? What exactly do we see? Are the cells in FP1 head to head and in FP2 side by side? What is the scale and orientation of the individual images? I will give more examples in the minor point section.

We are very grateful for this comment and we have made the presented images easier to interpret throughout the manuscript, including the inclusion of a schematic cartoon to help orientate the readers. We have also added scale bars to all images where they were absent, and included new data from cryosections of the memGFP OFM at regions of fusion and fused seam to provide greater clarity for tissue and cellular orientation.

It would be good if the data mentioned in subsection “Chick OFC was characterised by the breakdown of basement membranes, loss of epithelial morphology and localised apoptosis” and subsection “Netrin-1 is specifically and dynamically expressed in the fusing OFM” could be shown.

In subsection “Chick OFC was characterised by the breakdown of basement membranes, loss of epithelial morphology and localised apoptosis”, we have added brightfield microscopy data using sections cut at the nascently fused seam to the updated Figure 1. This data shows the absence of RPE pigmentation in the in the post-fused region. In subsection “Chick OFC was characterised by the breakdown of basement membranes, loss of epithelial morphology and localised apoptosis”, the human NTN1 expression data in the OFM is part of a larger study using RNAseq in the fissure margin to identify fissure specific genes. This manuscript is currently in preparation and should be available as a preprint soon.

In some cases, the citations seemed incorrect, e.g. subsection “Transcriptional profiling reveals genetic conservation between chick and human OFC”, Cho and Cepko (2006).

This reference should be to Peters and Cepko (2002) and not Cho and Cepko (2006). We apologise for this error and have corrected the reference. We have also checked the accuracy of other references throughout the manuscript.

Subsection “Transcriptional profiling reveals genetic conservation between chick and human OFC”: what would be the definition of "biologically significant" in terms of differential gene expression? Would this not be dependent on the gene and gene product? Can this be generalized?

We used fold change (FC) analysis to identify biologically-relevant differential gene expression (Log_2_FC ≥1.5 or ≤-1) in the fissure compared to whole eye. These are established levels to determine biologically relevant gene expression changes between treatments or groups. The “significance” we used applied a false discovery rate (FDR) adjusted p-value of < 0.05 to determine differential expression. However, we also made use of TPM values as a direct read of gene expression levels, choosing to pay more attention to those genes with observed of >12 transcripts observed per million reads. We included all exons for each gene and therefore were not able to determine alternative splicing events.

Subsection “Transcriptional profiling reveals genetic conservation between chick and human OFC”: Table 2 is not showing this. Figure 3B shows the 12 increased only.

Table 3 has been edited to more clearly highlight genes with increased or decreased expression values.

Figure 1A: Is the OFM really the unpigmented region? Could it be the fissure itself?

The region depicted as OFM in Figure 1A (HH.St25) is actually a combination of non-pigmented OFM and the intervening peri-ocular mesenchyme. This is more clearly illustrated in Figure 1—figure supplement 1.

Figure 1B: To what extent is the iris developed at the stage of fissure fusion?

On the request of the reviewer we have added a supplemental figure (Figure 1—figure supplement 2) showing the superficial development of both the iris and pecten with relevance to OFC progression. The chick iris is apparently unique in among vertebrate models of OFC in that the pupillary region (the anterior-most part of the OFM) fuses early during development (~HH.St26) but remains distinct from the main epithelial fusion events described in our work. Indeed, the region posterior to this fused area remains open throughout development and is abutted by the static FP1. This open region in the iris allows a blood vessel to enter the eye to nourish the pecten, as described in our initial Results section. Other than this, the iris does not play a direct role in OFC in the chick but future studies could investigate the mechanisms by which the development of FP1 is arrested at the iris region.

The pecten could be explained more. On the images, it appears as a gap.

We have added data showing the structural development of the pecten in Figure 1—figure supplement 2, and clearly indicate the pecten region in the schematic we have added to Figure 1. However, the recent report by Bernstein et al. illustrated pecten development in the proximal chick optic fissure and highlighted its distinct intercalation fusion mechanism. We did not observe any pecten involvement at the epithelial fusion in the medial and distal OFM. Very little is known about pecten function, and as humans and mammals do not have pecten we did not study its development further.

Could the axes be renamed to proximal and distal?

Yes. We have changed to use of proximal-distal axes throughout

Figure 1C: What is the orientation, magnification and how was the labelling achieved?

These are flat mount confocal microscopy single plane images of an OFM from the stable memGFP chick line and are now more clearly described in the legend. To aid interpretation, we have now added clearer P-D axis orientation labels and scale bars, and a schematic of the developing OFM in the context of the whole eye.

Figure 1E: Were this and Figure 1—figure supplement 1G independent datasets?

Yes. The section data was not used to quantitate PH3A in the developing OFM, rather it was used to orientate the dividing cells within the fissure. This data is now placed in Figure 2—figure supplement 1.

Figure 1F: Is the proliferation increased outside or reduced inside?

This is an excellent question. We were careful not to over-interpret this data but used it to confirm there are fewer proliferating cells within the seam than in the surrounding tissue, and that therefore cell-division is not a major mechanism for seam expansion.

Figure 1—figure supplement 1B: What is seen here? This image showed the pecten and blood vessel at the iris.

We have added to and clarified this data in a new stand-alone Figure 1—figure supplement 2, and improved the descriptions in the figure legends.

Figure 1—figure supplement 1C: Is this taken from flat mounted tissue? What is the orientation?

Yes, this is flat mounted tissue with P-D axis orientation. We have improved the annotation of this figure.

Figure 1—figure supplement 1D: It is stated that at HH26 there is no fusion plate. Could higher magnifications be provided? In Figure 1—figure supplement 1Di it is not totally clear, especially comparing it to Figure 2B.

We have provided increased magnification images of the iris region in Figure 1—figure supplement 1Di. This panel corresponds to the distal-most region of the iris and is adjacent to the iris OFM that remains open throughout chick eye development (see comments above).

Figure 1—figure supplement 1G: The label "G" is missing in the figure. I do not understand the note that at HH29 the seam length was too small to quantify. The length itself was quantified in Figure 1—figure supplement 1F.

The relevant label has been added. The grids used to quantitate PH3A positive-cells in the flat-mounted confocal analyses were too large (200 µm^2^) to include the small amount of fused seam in this region (100 µm). Nevertheless, the data at FP1 and FP2 included fused seam tissue and show fewer PH3A positive cells than in regions outwith the seam at this stage. Therefore, our data at both time points does include fused seam and shows a reduced amount of proliferating cells than tissue outwith the fused seams.

Table 1: Can an animal have FP1 and FP2? At stage HH29 n 5 were analyzed. 4 showed 2FP 5 showed 1FP. Could this be explained?

We observed subtle variability in the progression of fusion – see detailed response to similar comments by reviewer 1. We defined a single fusion plate as the presence of RPE and neuroepithelial continuum, and two fusion plates were reported where these were separated by visibly fused seam > 0.1mm. Two FPs, one proximal and one distal, were consistently visible during expansion of the fused seam during active fusion stages.

Figure 2:Without time-lapse analysis, the term dynamics is potentially misleading.

To reflect this comment, we have changed the title of Figure 2 to “Basement membrane remodelling, loss of epithelial characteristics and apoptosis are defining features of Chick OFC.”

Figure 2A: Arrowheads in fusion plate image point to a domain in which the basal lamina is still visible.

The arrowheads have been moved slightly to more accurately indicate the regions of BM dissolution.

Figure 2B: Can the authors be sure that the future NR and RPE are both contributing and are both remodeled? Could there be dynamic in the fissure margins?

We are confident cells from both origins are present in the fusion plate as we have observed both RPE and NR cells contributing to fusion in various experiments (e.g. Figure 2A, Figure 4A and Figure 4 [in human OFM], and in Figure 4—figure supplement 1). Intriguingly, the RPE cells in this region are unpigmented and both cell types lose their normal epithelial organisation. Both of the cell types also express NTN1 (See Figure 4A), suggesting that NTN1 has a role is specifying these cells for fusion, or temporarily prevents them from differentiating into NR or RPE cells until fusion has occurred. We are currently investigating the mechanisms of this process, and believe that it may be this specific role of NTN1 that is essential for normal fusion.

Figure 2C: The image presented to show the apposition is not corresponding to Figure 2B.

We consistently observed A-Casp-3 foci at FP2 in the pre-fusing tissue, FP and adjacently fused seam. We have repeated the cryosection A-Casp-3 immunofluorescence to provide updated and clearer panels for this figure.

Figure 2D: Which domain was used for quantification? What was considered seam and outside seam and what could be considered OFM and outside OFM (towards nasal and temporal directions respectively)?

We apologise this was not made clear in the original submission. The data shown were generated using a fixed-parameter region of interest approach to count apoptotic foci in serial cryosections at HH.St30. The serial sections were cut through the P-D axis and included >200 µm of unfused open fissure and fused seams in full. We have added the following passage to the Materials and methods section for the manuscript to make this clearer: “To quantitate apoptotic foci at the OFM, we used Activated-Casp3 immunofluorescence on serial cryosections of HH.St29-30 OFMs and collected confocal images for each section along the P-D axis. Image analysis was performed by counting A-Casp3 positive foci at the OFM in sequential sections using a region of interest of fixed dimensions of 100 µm^2^.” The ROI included NR and RPE, but not periocular mesenchyme. We also quantitated A-Casp3 foci in the dorsal, nasal and temporal retina away from the OFM, however we were unable to identify more than >1 foci per eye in many cases and therefore we did not present this data in the study.

Figure 2—figure supplement 1: What is meant by medium?

NF145 protein is also referred to as “neurofilament medium”.

What is the orientation? What is considered central retina?

We are grateful for the opportunity to improve this figure – we have amended this substantially to provide stronger data and to help interpret and orientate the reader, including a brightfield flat mounted OFM. Central retina is now clearly indicated in the schema and refers to a temporal region of optic cup.

Figure 3B: Could the boxes be separated for the different stages? The annotation over the boxed could be misunderstood for statistical data. The annotation (.) should be included in the legend. The legends mention CHDL1 and not CHRDL1.

We have amended the legend to correct these errors. The boxes in b refer to the data presented in Table 3, where the individual stage Log2FC data is shown. We feel the current representation sufficiently shows these genes are consistently enriched during all stages of OFC in the chick.

Figure 3—figure supplement 1A: The images are appreciated. Were the individual regions dissected manually and in independent eyes also for fissure and ventral eye, meaning, is ventral eye including the fissure?

Yes, all were dissected manually and were independent for all sample types. The ventral regions included the fissures to obtain graded expression data.

Figure 3—figure supplement 1B: How was the dorsal data acquired? Is the transcriptional data available?

The dorsal data was similarly segmentally dissected, and the data is available in the excel files supplied as Figure 3—source data 1 and Figure 3—source data 2.

Figure 4A-C: What is the orientation?

The new schematic we have included in the revised Figure 1 will help orientate all of the data in this manuscript and we have added P-D axes in this figure to provide additional clarity.

Figure 4B: What is meant by whole mount? At the FP2, the NTN1 signal seems less intense. Is the arrow positioned correctly? A co-staining with a basal lamina marker would be helpful to conclude the localization.

This is an error in the figure legend, and it should have read as “flat mount”. The tissue was processed as whole OFM for immunofluorescence as described in the methods section. The annotations were incorrectly positioned and have been amended. We agree with the reviewer however we have been unable to find a basal lamina marker that is compatible with the anti-NTN1 antibody in our immunofluorescence method.

Figure 4B’’: What is NT? Is the overall annotation correct? Is NTN1 expressed also in the POM?

This was an error in the figure legend. We have added additional data showing similar NTN1 localisation at FP1. We have also removed the section data as we have optimised the use of the NTN1 antibody for cryosections and have added this new data, which more clearly shows NTN1 localisation in the same regions as the original data. NTN1 was not identified in POM in any of our RNAscope, in situ hybridisation or immunofluorescence analyses.

Figure 4C, D: Higher magnifications are needed and potentially a co-labelling with a basal lamina marker.

In Figure 4C, these tissues were difficult to come by and unfortunately BM markers were not used. However, we have provided higher magnifications to help interpret the data and observe tissue architecture.

In Figure 4D, we have increased the magnification of the images to provide better interpretation of the data.

Figure 4F: What is shown in the second image?

The second image shows a Wild type mouse secondary palate with normal fusion. We have added the label “Palate” to the panels to make this clearer.

Figure 4—figure supplement 1B: The anti-laminin staining looks odd.

These images were taken from the anterior optic fissure margin in the iris region. We appreciate that the Laminin looks to be thicker than is typical and suggest that this is a result of generating a maximum image projection of a confocal Z-stack using sections cut at a slightly oblique angle. Nevertheless, the data in this figure clearly shows a reducing gradient of NTN1 mRNA in post-fused tissue and is representative of 3x biological replicates.

Figure 4—figure supplement 1C: What was the age of the embryos?

HHSt.30 – this information has now been added to the figure legend.

Arrows were used twice for different annotations.

We have updated the figure and now use yellow arrows to define the midline of the OFM in the P-D axis.

In the no antibody control, green signal can be seen. Was there a problem with color channel separation? It was stated that there was no staining.

When merging these raw images to create the composite image in FIJI, some green signal became visible in the NTN1 channel. However, it should be noted that in the single channel fluorescence image for the no-antibody control, there was clearly no signal except for some RBCs. In contrast, the NTN1 positive signal is strong and specifically localised in the samples where antibody was used. This data has been removed from the manuscript and has been replaced by immunofluorescence on cryosections (Figure 4C).

Figure 4—figure supplement 1D: Please add axes for orientation and landmarks for orientation.

We have now added the dorsal and ventral axes to this figure. This data is now in a new stand-alone figure: Figure 4—figure supplement 2.

Figure 4—figure supplement 1F: If the penetrance of a phenotype is given, could it be also supported by genotyping besides by calculation? Was the phenotype consistent with the image in f? The margins seem farther apart? Where the margins in "apposition" correctly? Histology should be performed, or confocal analysis, to address this issue. What is the age of the embryo presented?At the end of the legend there is a redundant text section.

This has been removed.

It would be good if the data mentioned in subsection “Chick OFC was characterised by the breakdown of basement membranes, loss of epithelial morphology and localised apoptosis” and subsection “Netrin-1 is specifically and dynamically expressed in the fusing OFM” could be also shown.

See previous comments to reviewer 1.

The RNAseq data was shared. It would be best if it was accessible also without too much bioinformatics background.

We are pleased to be able to share the RNAseq data as we believe this is an important resource to ocular and fusion biologists and avian geneticists. The analyses of RNAseq data for all stages is supplied as supplementary Excel files with simple ranked lists that can be downloaded and analysed easily. In addition, the raw RNAseq data will be available form publicly accessible database on publication: NCBI Gene Expression Omnibus database (http://www.ncbi.nlm.nih.gov/geo) – accession number GSE84916.